



# Model bias in simulating major chemical components of PM$_{2.5}$ in China

Ruqian Miao[1], Qi Chen[1,*], Yan Zheng[1], Xi Cheng[1], Yele Sun[2], Paul I. Palmer[3], Manish Shrivastava[4], Jianping Guo[5], Qiang Zhang[6], Yuhan Liu[1], Zhaofeng Tan[1, 7], Xuefei Ma[1], Shiyi Chen[1], Limin Zeng[1],

Keding Lu[1], Yuanhang Zhang[1]

[1]State Key Joint Laboratory of Environmental Simulation and Pollution Control, Beijing Innovation Center for Engineering Science and Advanced Technology, College of Environmental Sciences and Engineering, Peking University, Beijing, 100871, China
[2]State Key Laboratory of Atmospheric Boundary Layer Physics and Atmospheric Chemistry, Institute of Atmospheric Physics,

Chinese Academy of Sciences, Beijing, 100029, China
[3]School of GeoSciences, University of Edinburgh, Edinburgh, EH9 3FF, UK
[4]Pacific Northwest National Laboratory, Richland, Washington, 99352, USA
[5]State Key Laboratory of Severe Weather, Chinese Academy of Meteorological Sciences, Beijing, 100081, China
[6]Ministry of Education Key Laboratory for Earth System Modeling, Department of Earth System Science, Tsinghua University,

Beijing, 100084, China
[7]Institute of Energy and Climate Research, IEK-8: Troposphere, Forschungszentrum Jülich GmbH, Jülich, 52425, Germany

*Correspondence to: Qi Chen (qichenpku@pku.edu.cn)

**Abstract.** High concentrations of PM$_{2.5}$ (particulate matter with an aerodynamic diameter less than 2.5 μm) in China have caused severe visibility degradation. Accurate simulations of PM$_{2.5}$ and its chemical components are essential for evaluating

the effectiveness of pollution control strategies and the health and climate impacts of air pollution. In this study, we compared the GEOS-Chem model simulations with comprehensive data sets for organic aerosol (OA), sulfate, nitrate, and ammonium in China. Model results are evaluated spatially and temporally against observations. The new OA scheme with a simplified secondary organic aerosol (SOA) parameterization significantly improves the OA simulations in polluted urban areas. The model underestimates sulfate and overestimates nitrate for most of the sites throughout the year. More significant

underestimation of sulfate occurs in winter, while the overestimation of nitrate is extremely large in summer. Our model is unable to capture some of the main features in the diurnal pattern of the PM$_{2.5}$ chemical components, suggesting underrepresented processes. Potential model adjustments that may lead to a better representation of boundary layer height, precursor emissions, hydroxyl radical, heterogeneous formation of sulfate and nitrate, and the wet deposition of nitric acid and nitrate are tested in the sensitivity analysis. The results suggest that uncertainties in chemistry perhaps dominate the model

bias. The proper implementation of heterogeneous sulfate formation and the good estimates of the concentrations of sulfur dioxide and hydroxyl radical are essential for the improvement of the sulfate simulation. The update of the heterogeneous uptake coefficient of nitrogen dioxide significantly reduces the modeled concentrations of nitrate, and accurate sulfate simulation is important for modeling nitrate. However, the large overestimation of nitrate concentrations remains in summer for all tested cases. The uncertainty of the production of nitrate cannot explain the model overestimation, suggesting a problem

related to the removal. A better understanding of the atmospheric nitrogen budget is needed for future model studies. Moreover,





the results suggest that the remaining underestimation of OA in the model is associated with the underrepresented production of SOA.

## 1 Introduction

In developing countries like China and India, the concentrations of $PM_{2.5}$ (particulate matter with an aerodynamic diameter less than 2.5 μm) often exceed air-quality standards, leading to visibility reduction and negative health effects (Chan and Yao, 2008; Lelieveld et al., 2015). Chemical transport models (CTMs) are valuable tools to evaluate the $PM_{2.5}$ pollution and its health and climate impacts. Many studies have shown reasonable simulations of surface $PM_{2.5}$ concentrations in China by the CTMs. For example, the Weather Research and Forecasting/Community Multi-scale Air Quality (WRF/CMAQ) model has reproduced the monthly-averaged concentrations of $PM_{2.5}$ at the air-quality sites in 60 Chinese cities (J. Hu et al., 2016). The MICS-Asia Phase III studies further show the normalized mean biases (NMBs) of less than 50% for daily or monthly mean $PM_{2.5}$ concentrations in various CTMs (Gao et al., 2018; Chen et al., 2019). However, the model performance on $PM_{2.5}$ is component-dependent and may contain compensation errors, which bias the evaluation of the effectiveness of the emission control strategies. Recent model evaluations have reached an agreement that CTMs generally underestimate the concentrations of organic aerosol (OA) (Fu et al., 2012; Han et al., 2016) and sulfate (Wang et al., 2014; G. J. Zheng et al., 2015) but overestimate the concentrations of nitrate (Wang et al., 2013; Chen et al., 2019). During the severe haze periods, the models often significantly underestimate the $PM_{2.5}$ concentrations (Wang et al., 2014; G. J. Zheng et al., 2015).

Uncertainties exist in meteorological fields, emission inventories, and the physical and chemical processes, which contribute to the model biases in the $PM_{2.5}$ simulations. For example, models are well recognized to reproduce temperature (T) and relative humidity (RH), but are difficult to capture the near-surface wind fields (Guo et al., 2016a; Gao et al., 2018; J. Hu et al., 2016). Boundary layer structures greatly affect the $PM_{2.5}$ concentrations (Z. Li et al., 2017; Su et al., 2018). Evaluations of the boundary layer (e.g., boundary layer height (BLH)) in the CTMs are however limited (Bei et al., 2017; Chen et al., 2016). For typical primary components and secondary precursors of $PM_{2.5}$, the uncertainties of their emissions in Asia range from tens to several hundreds of percent (M. Li et al., 2017). The bottom-up and top-down estimates of the emissions of sulfur dioxide ($SO_2$), nitrogen oxides ($NO_x$), ammonia ($NH_3$), volatile organic compounds (VOCs) and organic carbon (OC) show significant differences in magnitude and seasonal variability (Koukouli et al., 2018; Qu et al., 2019; L. Zhang et al., 2018; Cao et al., 2018; Fu et al., 2012).

For sulfate, the model underestimation has been attributed largely to heterogeneous production. The proposed heterogeneous formation mechanisms include the $SO_2$ oxidation by nitrogen dioxide ($NO_2$) directly (Cheng et al., 2016; Wang et al., 2016) or indirectly (L. Li et al., 2018), by $O_2$ via transition-metal-ion (TMI) catalysis (G. Li et al., 2017) or radical chain reactions (Hung and Hoffmann, 2015; Hung et al., 2018), and by hydrogen peroxide (Ye et al., 2018). Among them, TMI-catalyzed oxidation of $SO_2$ perhaps dominates the sulfate formation during the haze periods, constrained by the observations of sulfate





oxygen isotopes (Shao et al., 2019). Although the mechanisms are under debate, the heterogeneous formation has been simplified in models as a reactive uptake process to achieve a better agreement of sulfate concentrations during the haze episodes (Wang et al., 2014; G. J. Zheng et al., 2015; J. Li et al., 2018). For nitrate, the contribution of heterogeneous chemistry remains largely uncertain. The uptake coefficients of dinitrogen pentoxide ($N_2O_5$), $NO_2$, and nitrate radical ($NO_3\cdot$) are sensitive to experimental conditions and range orders of magnitude (Bertram and Thornton, 2009; McDuffie et al., 2018; Brown and Stutz, 2012; Spataro and Ianniello, 2014). The parameterizations of heterogeneous production of nitrate differ significantly among models (Holmes et al., 2019; Alexander et al., 2020; J. Li et al., 2018; Wang et al., 2012). The simulations of sulfate and nitrate affect the simulation of ammonium through thermodynamic equilibrium. For OA, the complexity of its secondary formation and aging processes and the lack of emission inventories of intermediate-volatility (IVOCs) and semivolatile organic compounds (SVOCs) affect the model performance (Chen et al., 2017 and references therein). Substantial model-observation discrepancies present in the comparisons of the mass concentration and the oxidation state of OA as well as the contributions of various formation pathways (Tsigaridis et al., 2014; Heald et al., 2011; Chen et al., 2015). Moreover, oxidant levels affect the chemical processes (Lu et al., 2018). The model capability in simulating the concentrations of major oxidants like hydroxyl radical ($OH\cdot$) and hydroperoxy radical ($HO_2\cdot$) are rarely evaluated.

The net model bias caused by the above factors can be non-linear. Various factors may interact with each other and thus alter the model bias, which needs to be evaluated systematically. On the other hand, observations may be biased and contribute to the model-observation discrepancies. For example, filter-based analysis of the $PM_{2.5}$ components can contain positive or negative artifacts for semivolatile species, resulting from improper use of denuder and back-up filters (Liu et al., 2014). Such artifacts are often large (>50%) (Chow, 1995) and have been ignored in most model-observation comparisons (Wang et al., 2013; Qin et al., 2015). Online measurements by aerosol mass spectrometers have less uncertainty (~30%) compared to filter-based analysis (Canagaratna et al., 2007). Most of the measurements are however conducted for submicron particles (DeCarlo et al., 2006; Ng et al., 2011). The particulate mass in the supermicron domain needs to be considered in the model-observation comparisons (Elser et al., 2016).

In this study, we synthesized a comprehensive dataset of the concentrations of major $PM_{2.5}$ components (i.e., sulfate, nitrate, ammonium, and OA) from 55 online measurements at urban sites and 22 at non-urban sites in China. We evaluate the latest version of GEOS-Chem nested-grid model simulations with this dataset as well as a long-term online dataset that consists of hourly measurements of the major $PM_{2.5}$ components from 2011 to 2013 in Beijing. Potential factors that may contribute to the model-observation gaps are discussed. We also conducted sensitivity analysis for two case periods to show the potential contributions of different factors to the model-observation gaps and the contributions of various combinations of these factors.

## 2 Description of Observations

The campaign-average mass concentrations of sulfate, nitrate, ammonium, and OA as well as the sampling information are



listed in Table S1 of the Supporting Information (SI), including 77 surface online measurements from 2006 to 2016 in China. The dataset covers the regions of North China Plain (NCP), Yangtze River Delta (YRD), Pearl River Delta (PRD), and

100 Northwest China (NW). The measurements are made by Aerodyne high-resolution time-of-flight aerosol mass spectrometer (HR-ToF-AMS), quadrupole aerosol mass spectrometer (Q-AMS), and aerosol chemical speciation monitor (ACSM) and are mostly for submicron particles (Y. J. Li et al., 2017). We also compared our model simulations to long-term ACSM measurements of submicron particle composition at the site of Institute of Atmospheric Physics, Beijing (IAP, 39°58′28″ N, 116°22′16″ E) from July 2011 to May 2013 (Sun et al., 2015). The long-term data have a time resolution of 15 minutes and

105 were averaged to an hour when comparing with the model results. All data were corrected by collection efficiency as stated in the original publications. Our recent measurements show that the submicron-to-fine ratios for sulfate, nitrate, ammonium, and OA are quite similar (i.e., 0.8) for the summertime and wintertime measurement periods in Beijing except for the severe winter-haze episodes under high RH (i.e., about 0.5) (Fig. S1 in SI) (Zheng et al., 2020). For simplicity, we divided the observation data herein by 0.8 for the four species when comparing to the model results.

The meteorological parameters (e.g., T, RH, wind speed, and wind direction) and the concentrations of gaseous pollutants including ozone ($O_3$), carbon monoxide (CO), $SO_2$, and $NO_2$ were measured at the Peking University Urban-Atmosphere Environment Monitoring Station (PKUERS, 39°59′21″ N, 116°18′25″ E) from July 2011 to May 2013. Both the IAP and PKUERS sites are in the same GEOS-Chem model grid. The monthly mean $NH_3$ concentrations are taken from the 2007-2010 observations at the IAP site (Pan et al., 2012). The BLH in Beijing (39°48′00″ N, 116°28′12″ E) was derived from the

radiosonde observations at 8 AM, 2 PM (only in the summer), and 8 PM during July 2011 to May 2013 by using bulk Richardson algorithms (Guo et al., 2016b; Guo et al., 2019). All the hours refer to Beijing time (UTC+8). The radiosonde-derived BLH is greater in spring and summer and lower in autumn and winter, which is consistent with the findings from the satellite observations and the ground-based ceilometer measurements (W. Zhang et al., 2016; Tang et al., 2016).

Moreover, the observed concentrations of OH· and $HO_2$·, gaseous nitrous acid (HONO) and nitric acid ($HNO_3$), and isoprene

in Beijing are taken from literature, including the studies in south Beijing (Wangdu, 38°39′36″ N, 115°12′00″ E) from 7 June to 8 July 2014 and in north Beijing (Huairou, 40°24′36″ N, 116°40′48″ E) from 6 January to 5 March 2016 (Tan et al., 2017; Tan et al., 2018; Liu et al., 2019), and additional isoprene measurements at the PKUERS site during the summer of 2011 (Zhang et al., 2014). The observed concentrations of $NO_3$· and aromatic compounds are taken from the measurements at the PKUERS site in September 2016 and in summer and winter of 2011-2012 (Wang et al., 2015), respectively.

**3 Model Description**

The atmospheric chemical transport model GEOS-Chem 12.0.0 (http://geos-chem.org) was run at nested grids with 0.5°×0.625° horizontal resolution over Asia and adjacent area (11°S-55°N, 60°-150°E) and 47 vertical levels between the surface and ~0.01 hPa. Boundary conditions were provided by the global simulations at 2°×2.5° horizontal resolution. Both global and nested





simulations were spun up for one month. MERRA2 reanalysis meteorological data from the NASA Global Modeling and
Assimilation Office (GMAO) were used to drive the model. Model simulations were run for the measurement period of July
2011 to May 2013 to compare with long-term data sets. When comparing with the campaign-average data, the model
simulations for the year of 2012 were used. For other comparisons, the model simulations were run for the measurement
periods.

The GEOS-Chem model simulates the ozone-NOx-hydrocarbon-aerosol chemistry (Park et al., 2003; Park et al., 2004; Liao
et al., 2007). Aerosol thermodynamic equilibrium is performed by ISORROPIA-II (Fountoukis and Nenes, 2007; Pye et al.,
2009). The simulation of OA includes primary organic aerosol (POA) and secondary organic aerosol (SOA). The model
assumes that 50% of POA emitted from combustion sources are hydrophobic and hydrophobic POA converts to hydrophilic
POA with an e-folding time of 1.15 days. A ratio of 1.6 is applied to account for the non-carbon mass in POA (Turpin et al.,
2000). SOA is simulated by the Simple SOA scheme (Hodzic and Jimenez, 2011; Kim et al., 2015). SOA precursor surrogates
are estimated from the emissions of biogenic volatile organic compounds (i.e., isoprene and terpenes) and CO from the
combustion of biomass, biofuel, and fossil fuel. The Simple SOA scheme assumes that the irreversible conversion from
precursors to particle-phase SOA takes a fixed timescale of 1 day and that 50% of biogenic SOA precursors are emitted as
particle-phase SOA. The SOA yields of isoprene and terpenes are set to be 3% and 10%, respectively. The SOA yield of
biomass burning emissions is set to be 1.3% of CO, and the yield for fossil-fuel combustion is set to be 6.9%. These yields are
derived from the observed ratios between SOA and CO in aged air masses from the studies in the United States (US) (Hayes
et al., 2015) and are able to reproduce the OA mass without detailed SOA chemistry in the southeast US (Kim et al., 2015).
Because of the lack of related measurements in China, we did not change these yields herein.

Wet depositions of soluble aerosols and gases include convective updraft, rainout, and washout as described by Liu et al.
(2001). SOA is treated as highly soluble with a fixed Henry's law coefficient of $10^5$ M atm$^{-1}$ and a scavenging efficiency of
80% for simplicity (Chung and Seinfeld, 2002). The Henry's law coefficients may vary in magnitudes depending on the SOA
types (Hodzic et al., 2014). Hodzic et al. (2016) shows similar vertical profiles of modeled SOA mass for using the fixed $10^5$
M atm$^{-1}$ and the volatility-dependent Henry's law coefficients. Dry deposition is calculated by a standard resistance-in-series
model for the aerodynamic, boundary-layer, and canopy-surface resistance (Wesely, 1989).

Global anthropogenic emissions in GEOS-Chem are provided by the Community Emissions Data System (CEDS) (Hoesly et
al., 2018), including the monthly emissions of gaseous pollutants ($SO_2$, $NO_x$, $NH_3$, $CH_4$, CO, and VOCs) and carbonaceous
aerosols (black carbon (BC) and OC). Anthropogenic emissions of CO, $NO_x$, $SO_2$, BC, OC, and VOCs in China are provided
by the Multi-resolution Emission Inventory for China (MEIC v1.3; http://meicmodel.org) for the years of 2010, 2012, and
2014. The emissions in 2011 and 2013 are interpolated from the emissions of the two adjacent years. We use an improved
inventory for agriculture emissions of $NH_3$ in China (L. Zhang et al., 2018). This inventory shows stronger peak emissions in
the summer than other inventories such as the Regional Emission in Asia (REAS2), PKU-$NH_3$, and the Emission Database for
Global Atmospheric Research (EDGAR) show, which agrees better with the top-down estimates. The non-agricultural $NH_3$



emissions in China are taken from the study done by Huang et al. (2012), which is based on the year of 2006 and represents the low-end estimates as the emissions increased rapidly after 2006 (Kang et al., 2016; Meng et al., 2017). The MIX Asian emission inventories are used for the anthropogenic emissions in the rest part of Asia (M. Li et al., 2017), which has combined

the South Korea inventory (CAPSS) (Lee et al., 2011), the Indian inventory (ANL-India) (Lu et al., 2011; Lu and Streets, 2012) and the REAS2 inventory (Kurokawa et al., 2013). Our simulations used sector-specific MEIC diurnal patterns for the anthropogenic emissions of CO, $NO_x$, $SO_2$, BC, OC, and VOCs from power, industry, residential, transportation, and agriculture sectors (Fig. S2 in SI) and the MEIC agriculture diurnal patterns for all anthropogenic emissions of $NH_3$ in China. $NO_x$ emission from soils and lightning are included in the model (Hudman et al., 2012; Murray et al., 2012). The biogenic

emissions are calculated from the Model of Emissions of Gases and Aerosols from Nature (MEGAN v2.1) (Guenther et al., 2012). The emissions from biomass burning are provided by the Global Fire Emission Database (GFED4) (Giglio et al., 2013).

Heterogeneous uptake of $SO_2$ into aerosol liquid water is not included in the standard simulations but in the sensitivity runs in Sect. 4.3. The parameterizations of the $SO_2$ uptake coefficient ($\gamma_{SO2}$) include $\gamma_{SO2}$ depending on RH or on aerosol liquid water content (ALWC) (B. Zheng et al., 2015; J. Li et al., 2018). The heterogeneous uptake of $N_2O_5$ and $NO_2$ is an important

contributor to nitrate in northern China (Wang et al., 2017; Wen et al., 2018). The uptake coefficients of $\gamma_{N2O5}$ and $\gamma_{NO2}$ on the aerosol surface vary by several orders of magnitude, depending on temperature, particle particle-phase state, composition, ALWC, pH and so on (Bertram and Thornton, 2009; Abbatt et al., 2012; McDuffie et al., 2018 and references therein). The standard model uses relatively high values of $\gamma_{N2O5}$ and $\gamma_{NO2}$ (McDuffie et al., 2018; Davis et al., 2008). Lower values are tested in the sensitivity runs. For SOA, the heterogeneous formation varies by sources and aging processes (Donahue et al., 2012).

We did not include any heterogeneous production of SOA because of the lack of good parameterizations (Chen et al., 2017).

## 4 Results and Discussion

### 4.1 Compensating errors in simulating PM$_{2.5}$

Figure 1 shows the scatter plots of the simulated and the observed campaign-average concentrations of secondary inorganic aerosol (SIA) and OA over China as well as the statistical values such as NMB, root mean square error (RMSE), and Pearson's

correlation coefficient ($R$) for the model-observation comparisons. For most of the sites, the sulfate concentrations are underestimated (NMB = −0.39, $R$ = 0.45), while the nitrate concentrations are overestimated (NMB = 0.82, $R$ = 0.57) by the model. Such underestimation for sulfate and the overestimation for nitrate in CTMs is a known problem for China (Gao et al., 2018; Chen et al., 2019). The simulations of ammonium concentrations (NMB = 0.06, $R$ = 0.58) are better than the simulations for sulfate and nitrate. For OA, previous model studies typically underestimate the OA mass concentrations by over 40% (Fu

et al., 2012; Zhao et al., 2016). The model herein shows improved performance by using the Simple SOA scheme (NMB = −0.26, $R$ = 0.70). Moreover, the model biases vary by season. For example, the sulfate concentrations are mostly



underestimated in winter (Fig. 1*a*) when the SO₂ emissions are plausibly underestimated (Wang et al., 2014; Koukouli et al., 2018). The model-observation agreement for nitrate is the best in winter (Fig. 1*b*).

Tables S2, S3, and S4 in SI list the statistical values for the model-observation comparisons in different regions, urban or non-urban sites, and various seasons, respectively. The model biases for sulfate, nitrate, and OA are consistently positive or negative among regions (Table S2), suggesting that the model biases are general problems in China. The underestimation of sulfate is over 40% (NMB) in most regions except YRD, and the overestimation of nitrate is over 80% (NMB) in most regions except NW. The OA simulations show much lower NMB (−10%) and RMSE values in YRD and PRD than in NCP and NW. For ammonium, the model significantly overestimates its concentrations in YRD and underestimates its concentrations in NW. The former may be explained by the excessive formation of ammonium through thermodynamic equilibrium under conditions of abundant $NH_3$ emissions (L. Zhang et al., 2018) and overestimated nitrate concentrations (Wang et al., 2013) in YRD in the model. The latter is likely a result of combined factors including emissions, meteorology, and thermodynamic equilibrium. Moreover, the mean observed concentrations of sulfate, nitrate, ammonium, and OA at urban sites are 20-90% greater than those at non-urban sites (Fig. 1 and Table S3). The model also shows greater simulated concentrations of OA at urban sites, similar to the observations. The model-observation gaps for nitrate (NMB = 1.22) and ammonium (NMB = 0.33) are greater in non-urban areas, whereas the gaps for sulfate (NMB = −0.44) and OA (NMB = −0.31) are greater in urban areas. This result suggests perhaps different driving forces of the model biases for the SIA species.

For seasonal variations, the underestimation of sulfate occurs all year round, and the greatest underestimation occurs in winter (NMB = −0.54) (Table S4). Similar to other models, our model failed to reproduce the high sulfate concentrations during the winter-haze periods (Wang et al., 2014; G. J. Zheng et al., 2015). By contrast, the nitrate concentrations are largely overestimated in spring, summer, and autumn (NMB = 0.79-1.28). The model bias is much smaller in winter (NMB = 0.41) when higher concentrations of nitrate present. Wang et al. (2013) also showed the summertime overestimation for East Asia. Heald et al. (2012) showed the summer-, autumn-, and winter-time overestimation for the eastern US. To sum up, the large overestimation of nitrate happens in most seasons and regions, and is more severe in non-urban sites. For ammonium, the model underestimates its concentrations in winter and spring but overestimates its concentrations in summer and autumn. Both the uncertainties of $HNO_3$ and $NH_3$ simulations may affect the modeled ammonium concentrations (Wen et al., 2018; Xu et al., 2019). The underestimation of OA is another year-round problem, and the worst case happens in autumn. The *R* value however is much lower in summer (i.e., 0.28 compared with ≥ 0.5 in other seasons), showing the complexity of the OA simulations.

The model simulations are further compared to the long-term hourly observations in Beijing. Figure S3 in SI shows the simulation-to-observation ratios for the SIA species and OA. The mean and median values of the simulation-to-observation ratios of the mass concentrations of non-refractory PM₂.₅ (NR-PM₂.₅) are generally within the measurement uncertainty of 30%. Compensation of the underestimation of sulfate and OA and the overestimation of nitrate leads to the good performance on NR-PM₂.₅. The seasonal variations of the model biases for the SIA species and OA in Beijing are consistent with the findings





in the nation-wide comparisons (Table S4), except that the greatest underestimation of OA occurs in spring instead of autumn. Figure S4 in SI shows the simulation-to-observation ratios when excluding the periods of NR-PM$_{2.5}$ mass concentrations over 150 µg m$^{-3}$. The model biases and their seasonal variations are similar to those in Fig. S3.

Figure 2 shows the diurnal patterns of the observed and the simulated concentrations of sulfate, nitrate, ammonium, and OA for four seasons in Beijing. Considerable differences exist. For instance, the observed sulfate shows a daytime concentration

build-up in spring and summer, suggesting a photochemical production (Sun et al., 2015). The wintertime diurnal pattern shows a steady but later enhancement in the afternoon. The simulated profiles however show insignificant daytime concentrations elevations in the model, suggesting insufficient production, overestimated boundary-layer dilution, or removal during the day (Fig. 2$a$). By contrast, the observed nitrate and ammonium concentrations show flatter diurnal patterns than sulfate (Fig. 2$b$-$c$). Nighttime production of nitrate by the heterogeneous uptake of N$_2$O$_5$ and NO$_2$ is a major pathway of nitrate

production in northern China (Wang et al., 2018; Alexander et al., 2020). The 2-5 times greater concentrations of simulated nitrate at night suggest overpredicted nighttime production, underestimated boundary-layer dilution, or underestimated removal of nitrate. On the other hand, the simulated profiles for nitrate and ammonium show large reductions of daytime concentrations especially in summer, which are not shown in the observed profiles. This may suggest insufficient daytime production of nitrate, overestimated daytime boundary-layer dilution or removal, or overestimated evaporation of ammonium

nitrate in the model. It is unclear whether the model treats the thermodynamics of evaporation properly when particles are coated with OA (Li et al., 2016). The model largely overestimates gaseous HNO$_3$ concentrations in Beijing (Fig. S5 in SI). If the evaporation of ammonium nitrate is limited because of the coating, the daytime variations of ammonium nitrate can be flatter and the daytime HNO$_3$ concentrations may be lower.

For OA, the model is unable to reproduce the midday and evening peaks for all seasons (Fig. 2$d$). Previous positive matrix

factorization (PMF) analysis of the OA mass spectra suggests that cooking emissions contribute to the midday peaks of the OA concentrations and the evening peaks are driven by mixed primary emissions including cooking, traffic, and coal combustion (W. Hu et al., 2016; Sun et al., 2015). Cooking emissions are not included explicitly in the model, and the emissions of POA and SOA precursors from traffic and coal combustion are uncertain (Tao et al., 2018; Peng et al., 2019). We compared the modeled POA and SOA with PMF-derived POA and oxygenated OA (OOA) (Sun et al., 2018). The model reproduces the

monthly mean concentrations of PMF-derived POA (Fig. 3$a$), suggesting that the MEIC POA inventory generally represents the particle-phase SVOCs emissions under ambient conditions. The model underestimation of OA is mainly from SOA as indicated by the underestimation of the monthly mean concentrations of PMF-derived OOA (i.e., 50-70% of the observed OA mass) (Fig. 3$b$). Figures S6 and S7 in SI show the model performance of the Simple SOA scheme and the traditional scheme (so-called Semivolatile POA scheme in GEOS-Chem) in simulating OA. The Semivolatile POA scheme significantly

underestimates both POA and SOA. This scheme treats 1.27 times of the POA inventory as the SVOC emissions, among which only 1.5% of the carbon remains as POA (Pye and Seinfeld, 2010). There is also a lack of constraints on the SOA production from IVOCs and SVOCs.



### 4.2 Potential contributors to the model-observation discrepancies

We focus here on measurements in Beijing to discuss about the potential contributors to the model bias. Table 1 lists the
statistics of T, RH, wind speed, wind direction, and BLH between the MERRA2 outputs and the observations in Beijing. The
MERRA2 reanalysis reproduces T (NMB < 2%) and RH (NMB < 15% except for winter) but is unable to reproduce the wind
speed and directions. Large RMSE for surface wind directions is a common problem in meteorological reanalysis products as
well as the WRF simulations. The overestimation of wind speed (1-2 times) is slightly greater than the bias reported in other
studies and may cause some underestimation of PM$_{2.5}$ (J. Hu et al., 2016; Wang et al., 2014). The MERRA2 slightly
overestimates 2 PM BLH compared with the radiosonde measurements in summer (NMB = 0.34). For 8 AM and 8 PM,
MERRA2 underestimates the radiosonde-derived BLH in autumn and winter. Bei et al. (2017) indicated that the uncertainty
in temperature and wind field simulations leads to the frequent underestimation of the nighttime BLH in January 2014 in
Beijing by the ensemble WRF meteorology. Such underestimation of BLH may lead to overestimated nighttime concentrations
of PM$_{2.5}$ in autumn and winter. The large RMSE values for the BLH comparisons at 8 AM and 8 PM suggest that the nighttime
simulation of PM$_{2.5}$ may have greater meteorological uncertainty than the daytime simulation (Fig. S8 and Table 1).

The emission inventories of SIA and SOA precursors are important model inputs (Huang et al., 2014). The uncertainty of SO$_2$
emissions affects surface sulfate concentrations. Our model underestimates SO$_2$ concentrations in winter and overestimates its
concentrations in summer in Beijing (Fig. 4$a$). Consistently, top-down estimates suggest lower SO$_2$ emissions in summer and
higher in winter in China compared with the MEIC inventory (Koukouli et al., 2018). Improving SO$_2$ emissions may reduce
the model bias for sulfate. Our model largely underestimates NO$_2$ concentrations year round (Fig. 4$b$). The bottom-up NO$_x$
inventory has about 50% of uncertainty (M. Li et al., 2017). Top-down estimates suggest lower NO$_2$ emissions in Beijing and
its surrounding area than the MEIC inventory (Qu et al., 2017). Moreover, laboratory and field measurements show that the
NO$_2$ uptake coefficient ($\gamma_{NO2}$) on the aerosol surface ranges from $10^{-8}$ to $10^{-4}$ (Spataro and Ianniello, 2014 and references therein;
M. Li et al., 2019 and references therein). The default GEOS-Chem model uses a relatively high $\gamma_{NO2}$ of $10^{-4}$, which may cause
the underestimated NO$_2$ concentrations as well as the overestimated concentrations of HNO$_3$, HONO, and nitrate and needs
further evaluation (Alexander et al., 2020). For NH$_3$, the model underestimates its monthly mean concentrations in Beijing
(Fig. 4$c$). The non-agriculture NH$_3$ emissions are based on the year of 2006 and can be greater in 2012 because of the rapid
economic growth (Kang et al., 2016; Meng et al., 2017). Several studies show that the non-agriculture emissions are the
dominant NH$_3$ sources during haze periods in Beijing (Pan et al., 2016; Sun et al., 2017). The underestimation of NH$_3$ affects
the ammonium simulations when the thermodynamic equilibrium is limited by gaseous NH$_3$. For SOA precursors, Fig. 4$d$
shows that the model underestimates surface CO concentrations in Beijing, which may contribute to the model underestimation
of anthropogenic SOA. The modeled summertime isoprene concentrations in Beijing are lower than the observations by 20-
50%, affecting the simulations of biogenic SOA (Table S5 in SI). The model also underestimates the aromatic concentrations,
similar to previous studies (Liu et al., 2012). However, such underestimation has little influence on SOA herein because that
aromatic SOA is modeled by the parameterization on CO in the Simple SOA scheme as part of anthropogenic SOA.



Oxidants are essential to chemical conversion. Figure 5*a-b* shows the modeled and the observed concentrations of OH· and HO$_2$· radicals in Beijing. The peak concentrations of OH· and HO$_2$· radicals are underestimated by a factor of 1.5-2 and 2-4, respectively, explained by the missing source of daytime HONO (Fig. S9) (Liu et al., 2019; L. Zhang et al., 2016; J. Zhang et al., 2018). Such underestimation suggests insufficient atmospheric oxidation capacity in the model, meaning reduced formation

of sulfate and nitrate. Figure 5*c* shows that the model overestimates the surface O$_3$ concentrations in winter. Common problems have been reported in other studies in China and other northern hemisphere places by various CTMs (J. Hu et al., 2016; Travis et al., 2016; Young et al., 2018; J. Li et al., 2019). Nevertheless, the overestimated O$_3$ has little influence on the SOA simulation by the Simple SOA scheme and has minor impacts on SIA because of the dominant contribution from the photochemical and heterogeneous pathways. Moreover, NO$_3$· affects the formation of nitrate and SOA (Ng et al., 2017). Measurements of NO$_3$· in

Beijing shows nighttime peak concentrations of less than 6 pptv in summer and below the detection limit of 2.7 pptv in winter. The modeled concentrations are three times greater than the peak concentrations in summer (Fig. 5*d*), suggesting a possible overestimation of nighttime oxidation.

In addition, the heterogeneous production of sulfate and SOA are not included in the standard models, leading to underestimations. The model uses relatively high values of $\gamma_{N2O5}$ and ignores the formation of nitryl chloride from the N$_2$O$_5$

uptake, both leading to the overestimation of nitrate (McDuffie et al., 2018; Davis et al., 2008; Jaegle et al., 2018). Another bias is the high default value of $\gamma_{NO2}$ as described previously. Biases may also relate to the atmospheric removal of the SIA species. For example, the GEOS-Chem model underestimates the wet deposition of nitrate in China by 15-23%, especially in urban areas in summer, which may affect both nitrate and ammonium (Zhao et al., 2017; Xu et al., 2018; Jaegle et al., 2018; Luo et al., 2019). The surface resistance of HNO$_3$ is overestimated in the model (Shah et al., 2018), although the test with

doubling the deposition velocity of HNO$_3$ suggests a minor impact of this factor on nitrate simulations (Heald et al., 2012). The photolysis rate of particle-phase nitrate affects the loss of nitrate (Romer et al., 2018; Kasibhatla et al., 2018). In Beijing, particulate nitrate may have lower photolysis rates because of the high mass concentrations and thick coating of PM$_{2.5}$ (Ye et al., 2017).

### 4.3 Relative importance of various factors to the model bias

For the various factors described above, we chose the ones that are expected to significantly affect the PM$_{2.5}$ simulations and can be constrained by ambient or laboratory measurements to conduct the sensitivity analysis. Their potential contributions to the model bias of PM$_{2.5}$ components are evaluated for two case periods, 21-26 August 2012 and 21-27 December 2012 for summer and winter, respectively. The simulations for both weeks show the highest $R$ value for the correlations with the observations during the seasons. Both of the weeks are absent from severe haze episodes during which unusual biases of the

meteorological fields and chemical processes may occur.

The tested factors for the sensitivity runs are listed in Table 2. Case 0 represents the standard model simulations. The nighttime BLH was multiplied by 3.6 based on the lowest median value of the MERRA2-to-observation ratios at 8 AM and 8 PM (Fig.


S8) when the original BLH was lower than 500 m (i.e., the median of the observed BLH) in Case 1. The $SO_2$ emissions in China were multiplied by 0.8 in summer and 1.5 in winter in Case 2 based on the minimum and maximum values of the ratios

between the top-down estimates provided by Koukouli et al. (2018) and the MEIC inventory (Fig. S10), respectively. The non-agriculture $NH_3$ emissions in China were scaled up by 1.4 as suggested by Kang et al. (2016) in Case 3. In Case 4, the reaction rate coefficients for the reactions that directly involve OH· oxidation and affect the formation and loss of $PM_{2.5}$ such as the gaseous formation of sulfuric acid and $HNO_3$ and the oxidation of $HNO_3$ were multiplied by 1.5 in summer and 2 in winter to offset the influence of underestimated OH· concentrations. The multipliers of 1.5 and 2 were derived on the basis of the largest

ratio of simulated to observed hourly mean OH· concentrations between 9 AM to 3 PM. In terms of the heterogeneous formation of sulfate, we added two types of parameterizations for $\gamma_{SO2}$ in Cases 5 and 6. One derives the uptake coefficient of $SO_2$ from RH ($\gamma_{SO2-RH}$) (B. Zheng et al., 2015), and the other calculates the coefficient as a function of ALWC ($\gamma_{SO2-ALWC}$) (J. Li et al., 2018). The former is in the order of $10^{-5}$, and the latter is in the range of $10^{-6}$ to $10^{-4}$. For comparisons, the uptake coefficients are $5 \times 10^{-5}$ in G. Li et al. (2017) and $10^{-9}$ to $10^{-3}$ in Shao et al. (2019). In Case 7, we reduced the value of $\gamma_{N2O5}$ from

the parameterization of Evans and Jacob (2005) to $10^{-3}$ to represent the lower end in the world-wide observations (McDuffie et al., 2018 and references therein). Similarly, the $\gamma_{NO2}$ was modified from $10^{-4}$ to $10^{-6}$ in Case 8 according to the median value of recent laboratory results (Spataro and Ianniello, 2014 and references therein; M. Li et al., 2019 and references therein). The wet deposition of nitrate is tested, for which we applied the seasonal variation to in-cloud condensation water for rainout parameterization and updated the washout parameterization of $HNO_3$ based on the method introduced by Luo et al. (2019) in

Case 9. Cases 10 to 50 are the runs with various combinations of the modifications in Cases 1 to 9 (Table S6 in SI) for the two case periods. We did not test any parameter related to OA because of the lack of sufficient ambient and laboratory constraints.

Figure 6 shows the simulation-to-observation ratios of hourly mean mass concentrations of NR-$PM_{2.5}$, sulfate, nitrate, and ammonium for Cases 0 to 9. The nocturnal BLH, the non-agriculture $NH_3$ emissions, the OH· levels, and the wet deposition of nitrate have minor impacts on the model performance of these components. The updated $SO_2$ emissions in Case 2 can

significantly improve the model simulation of sulfate in Beijing, although further improvements are needed in winter. Similar to previous findings, the heterogeneous uptake of $SO_2$ in Case 5 and 6 increases the simulated sulfate concentrations and leads to better model-observation comparisons in winter (B. Zheng et al., 2015; J. Li et al., 2018). However, both of the cases lead to the overestimation of sulfate concentrations in summer. The variances of the simulation-to-observation ratios for both cases are also greater than the standard simulation in Case 0, indicating the limitation of those parameterizations. Mechanistic

approach other than using indirect indicators like RH and ALWC may be necessary to improve the seasonality of the sulfate simulation.

The reduced $\gamma_{N2O5}$ in Case 7 leads to a minor reduction of simulated nitrate concentrations, suggesting that the uncertainty of heterogeneous uptake of $N_2O_5$ is not the main cause of the overestimation of nitrate. The simulations with more reasonable $\gamma_{NO2}$ in Case 8 are able to reproduce the observed nitrate concentrations in winter, indicating that biased $NO_2$ uptake is an

important contributor to the overestimation of nitrate. However, the updated $\gamma_{NO2}$ alone is insufficient to correct the nitrate





concentrations in summer, suggesting additional factors that contribute to the summertime overestimation. Given that the model overestimates both of the summertime concentrations of $HNO_3$ (Fig. S5) and nitrate, the bias is perhaps related to the insufficient removal of them. The updated wet deposition of nitrate can reduce the summertime monthly mean concentrations by about 20% (Fig. S11) but is still minor in terms of the large overestimation. Greater dry deposition of $HNO_3$ and faster

photolysis of particulate nitrate as well as the joint influence of multiple factors (as discussed later) are possible ways to solve the remaining overestimation.

Sulfate and nitrate simulations interact with each other through thermodynamic equilibrium, especially in winter when $NH_3$ emissions are lower than in summer. As shown in Fig. 6, adding the heterogeneous formation of sulfate reduces the simulation-to-observation ratios of nitrate in winter (i.e., the median ratio from 2.6 to 1.8-2.3 in Cases 5-6) and the simulated weekly mean

concentrations of nitrate by 16-36% (Fig. S12$a$). On the other hand, the reduced $\gamma_{NO2}$ leads to the reduction of the simulation-to-observation ratios of sulfate (i.e. about 0.1 reduction of the median ratios) and the weekly mean simulated sulfate concentrations by 12-20% (Fig. S12$b$). The reduced $\gamma_{NO2}$ decreases the HONO concentrations by 98% and hence the OH· levels by 26-74% in Beijing, which leads to lower concentrations of sulfate.

Figure 7 shows the $R$ and absolute NMB (|NMB|) values of the sulfate and nitrate simulations for Case 0, and Cases 5, 6, and

8 (i.e., updated heterogeneous formation), and Cases 10 to 50. In winter, the parameterization of heterogeneous sulfate formation on RH in Case 5 improves $R$ but leads to greater |NMB|, while the parameterization on ALWC in Case 6 leads to near-zero |NMB| but little changes of $R$. By contrast, Case 5 leads to worse values of both $R$ and |NMB|, and Case 6 only affects $R$ in summer. The results suggest that the parameterization on ALWC seems to be better in terms of overall model performance than the parameterization on RH. The decreased $R$ in summer in Case 6 is perhaps because that the biased

inorganic aerosol concentrations and the underrepresented organic contribution in the ALWC calculations lead to large uncertainty in the estimated ALWC and sulfate concentrations (Pye et al., 2009). For nitrate, the change of $\gamma_{NO2}$ in Case 8 leads to large improvements of either $R$ or NMB in both seasons.

The combination of the heterogeneous factors with other factors in Cases 10-50 shows various model improvements. For example, the combination of factors related to heterogeneous formation of sulfate, $SO_2$ emissions, OH· levels, and $\gamma_{NO2}$ shows

worse $R$ or |NMB| compared to Case 6 in winter (Fig. 7$a$) but improved model performance in summer (Fig. 7$b$). Such interaction suggests that the parameterization of heterogeneous sulfate formation is sensitive to the precursor concentrations and the oxidation conditions. Therefore, accurate $SO_2$ emissions and well-reproduced oxidant conditions are necessary for improving the sulfate simulation. For nitrate, the combinations of the $\gamma_{NO2}$ factor with other factors can worsen $R$ and |NMB| in winter. In particular, the combination of the improved $\gamma_{NO2}$ with the implementation of heterogeneous sulfate formation and

the updated $SO_2$ emission lead to the greatest reduction of $R$ and increase of |NMB| among cases (Fig. 7$c$), explained by the limitation of $NH_3$ relative to high sulfate concentrations. This impact is perhaps smaller in summer because of the greater $NH_3$ emissions. Accurate sulfate simulation therefore is important for the improvement of the simulation of wintertime nitrate in Beijing. The combination of various factors with the improved $\gamma_{NO2}$ leads to the consistent reduction of |NMB| in summer (Fig.





7*d*). Case 50 represents the combination of all factors (including $\gamma_{SO2\text{-}ALWC}$ not $\gamma_{SO2\text{-}RH}$). It shows an *R* value of 0.8/0.9

(winter/summer) and an |NMB| value of 0.05/0.3 for sulfate, and an *R* value of 0.8/0.7 and an |NMB| value of 0.3/2.1 for nitrate. By contrast, the standard simulation in Case 0 shows an *R* value of 0.9/0.9 and an |NMB| value of 0.6/0.3 for sulfate, and an *R* value of 0.9/0.7 and an |NMB| value of 2.0/4.7 for nitrate. For sulfate, the |NMB| is largely improved in winter by the combination of all factors. In summer, the influence of all factors seems being canceled out and therefore leads to an insignificant change in |NMB|. For nitrate, the combination of all factors can greatly improve the |NMB| in both seasons,

although the overestimation of nitrate is still very large in summer.

## 5 Conclusions

We evaluated the GEOS-Chem model simulations with a national-wide dataset in China and a long-term hourly dataset in Beijing for sulfate, nitrate, ammonium, and OA. The underestimation of sulfate and the overestimation of nitrate concentrations for most of the sites are consistent with previous findings. The Simple SOA scheme significantly improves the OA simulations

in China, suggesting that the SOA formation from anthropogenic precursors is perhaps the main reason for the underestimation of OA in previous studies. The model-observation agreement shows significant seasonality. Sulfate is mostly underestimated in winter, and nitrate is significantly overestimated except in winter. Our model is unable to reproduce the diurnal patterns of nitrate and ammonium. Sensitivity analysis for factors related to meteorology, emission, chemistry, and atmospheric removal with laboratory constraints show that uncertainties in chemistry perhaps dominate the model bias. Among the various

individual factors, updated heterogeneous parameterizations of $SO_2$ and $NO_2$ efficiently reduce the model-observation gaps of sulfate and nitrate, respectively. The impacts of various factors on model improvements are canceled out in some cases. Overall, the combination of all factors significantly improves the simulation for sulfate and nitrate. Because of the joint influence among factors, accurate $SO_2$ emissions as well as well-reproduced oxidant conditions and heterogeneous formation are essential for accurate sulfate simulation. Good sulfate simulation improves the nitrate simulation in urban areas with high anthropogenic

emissions. Mechanistic approaches other than parameterization on RH and ALWC are needed to improve the seasonality of the sulfate simulation. The summertime overestimation of nitrate remains the biggest problem in the model, which requires a better understanding of the atmospheric nitrogen budget. Simultaneous measurements of major reactive nitrogen species including $NO_x$, $N_2O_5$, $NO_3\cdot$, HONO, $HNO_3$, $NH_3$, and particle-phase nitrogen in the field campaigns can provide critical data sets for future model investigations. For OA, the remaining underestimation is plausibly associated with the insufficient SOA

production in the model, which merits further explicit investigations.

*Data availability*. Data presented in this manuscript are available upon request to the corresponding author.



*Author contributions.* QC designed the study. RM performed the model simulations and conducted the data analysis. YS, JG,
KL, YZ, SC, LZ, YZ, XC, YL, ZT, and XM provided the observation data. QZ provided the MEIC inventories and the diurnal
profiles of emissions. QC and RM prepared the manuscript with contributions from PIP, MS, JG, and KL.

*Competing interests.* The authors declare that they have no conflict of interest.

*Acknowledgments.* This work was supported by the MOST National Key R&D Program of China (2017YFC0209802), the
National Natural Science Foundation of China (51861135102, 91544107, 41875165). PIP was supported by NERC grant
#NE/N006879/1. This work was also supported by a special fund from the State Key Joint Laboratory of Environment
Simulation and Pollution Control (15Y02ESPCP and 16Y01ESPCP). The authors thank Weili Lin and Lin Zhang for technical
support and helpful discussion.

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





**Figure 1. Scatter plots of the simulated and observed campaign-average mass concentrations of (a) sulfate, (b) nitrate, (c) ammonium, and (d) OA in China. The solid and open symbols represent the urban and non-urban sites, respectively. Colors and shapes of the symbols represent seasons and regions, respectively. The observations were conducted during 2006 to 2016 for submicron particles and the data were divided by a submicron-to-fine ratio of 0.8. The model simulations were run for the year of 2012.**





**Figure 2.** Diurnal profiles of the simulated and observed hourly mean concentrations of (a) sulfate, (b) nitrate, (c) ammonium, and (d) OA at the IAP site in Beijing from July 2011 to May 2013. The observed concentrations were divided by a submicron-to-fine ratio of 0.8.

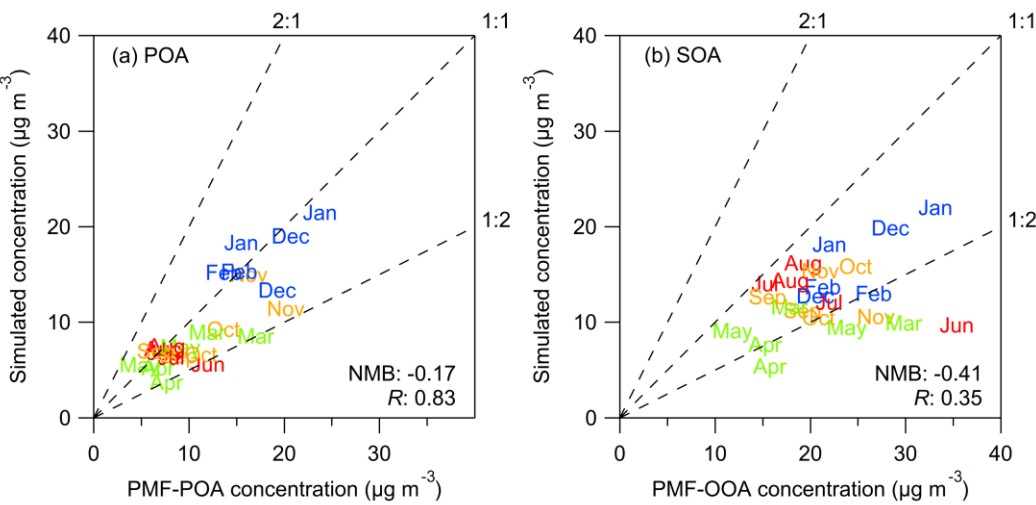

**Figure 3. Scatter plots of the monthly mean concentrations of (a) simulated POA and PMF-derived POA and (b) simulated SOA and PMF-derived OOA at the IAP site in Beijing from July 2011 to May 2013. The observed concentrations were divided by a submicron-to-fine ratio of 0.8.**

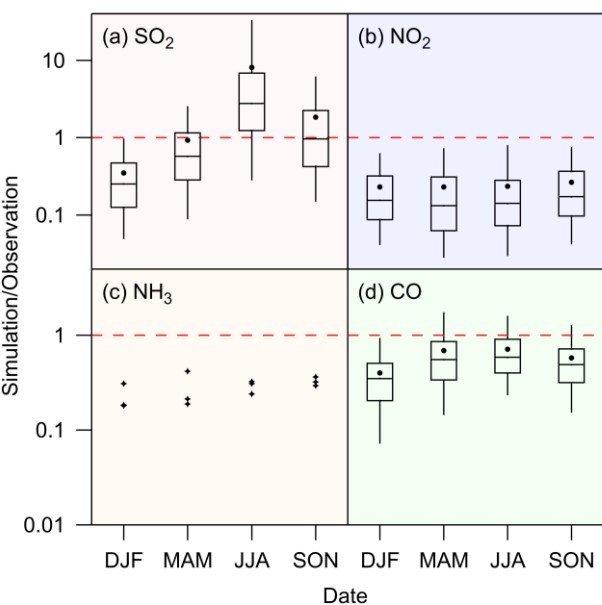

**Figure 4. The simulation-to-observation ratios of the concentrations of (a) SO₂, (b) NO₂, (c) NH₃, and (d) CO in Beijing. The upper and lower edges of the boxes, the whiskers, the middle lines, and the solid dots in panels a, b, and d denote the 25th and 75th percentiles, the 5th and 95th percentiles, the median values, and the mean values of the simulation-to-observation ratios of the hourly mean concentrations of the corresponding species at the PKUERS site from July 2011 to May 2013, respectively. The solid dots in panel c represent the simulation-to-observation ratios of the monthly mean concentrations of NH₃ at the IAP site from December 2007 to November 2010.**



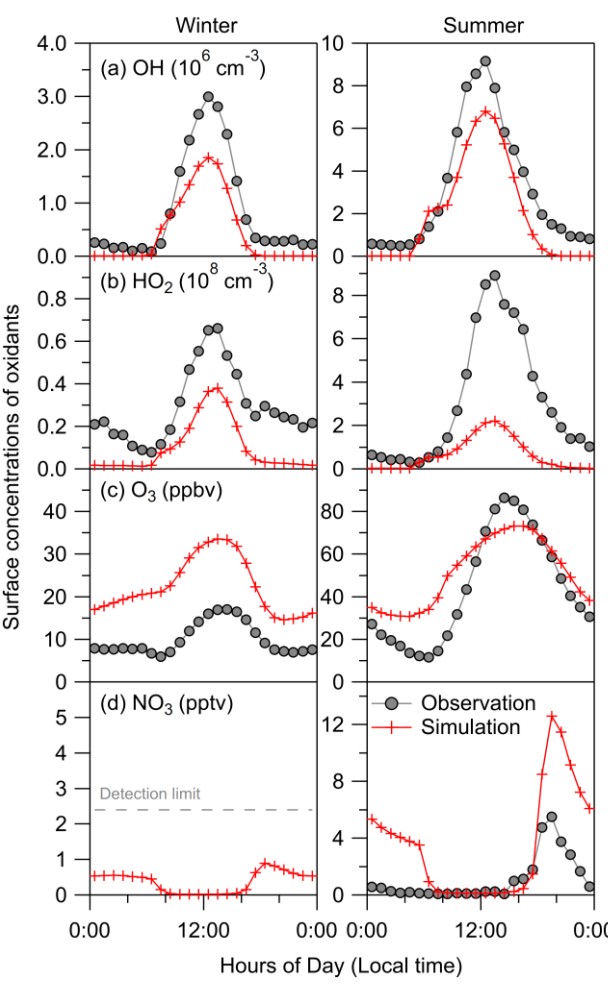

**Figure 5. Diurnal profiles of the hourly-mean simulated and observed concentrations of (a) OH and (b) HO₂ radicals at the Wangdu site from June to July 2014 and at the Huairou site from January to March 2016, (c) O₃ at the PKUERS site from July 2011 to May 2013, and (d) NO₃ radicals at the PKUERS site in September 2016 in Beijing.**



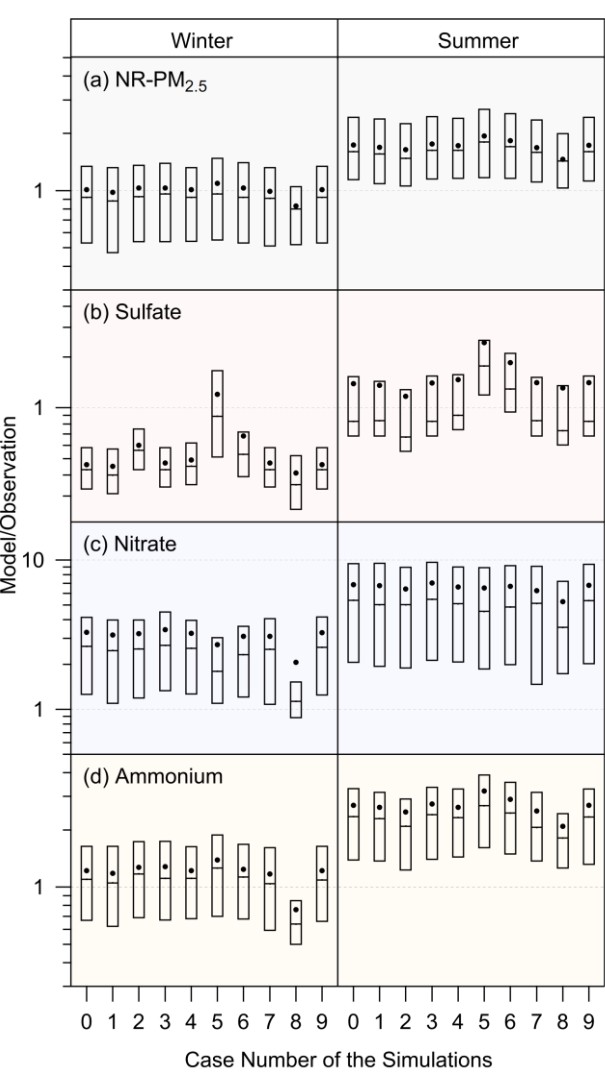

**Figure 6. Box and whisker plots of the simulation-to-observation ratios of hourly mean mass concentrations of (a) NR-PM$_{2.5}$, (b) sulfate, (c) nitrate, and (d) ammonium for the standard simulation (i.e., Case 0) and Cases 1 to 9 during the selected wintertime and summertime periods. The upper and lower edges of the boxes, the middle lines, and the solid dots denote the 25th and 75th percentiles, the median values, and the mean values, respectively.**



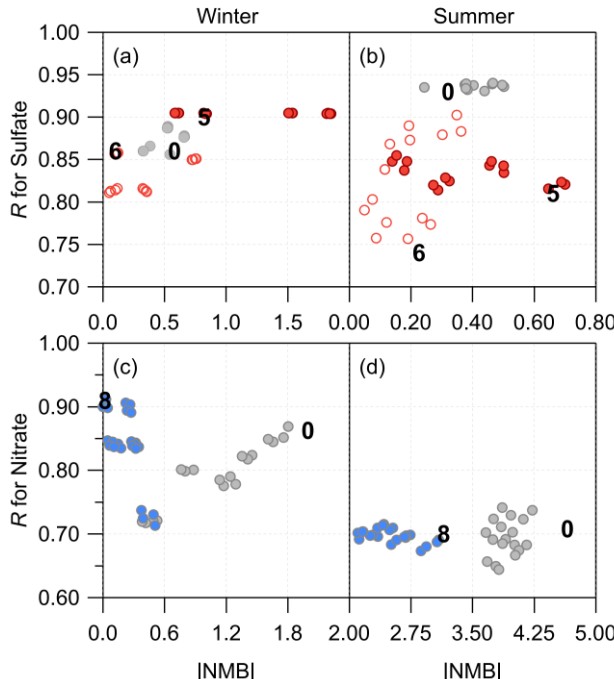

**Figure 7. Scatter plots of the _R_ and the absolute NMB values derived from the comparisons of the simulated and observed concentrations of (a-b) sulfate and (c-d) nitrate during the selected wintertime and summertime periods in Cases 10-50. For comparisons, Cases 0 (standard), 5 (implement $\gamma_{SO2\text{-}RH}$), 6 (implement $\gamma_{SO2\text{-}ALWC}$), and 8 (reduce $\gamma_{NO2}$) (Table 2) are also shown (marked as numbers). The solid red, open red, solid blue circles represent the cases that combine Cases 5, 6, and 8 with other changes, respectively, whereas the solid gray circles represent the rest of the cases (Table S6).**





**Table 1. Comparisons of the observed and simulated meteorological parameters, including T, RH, wind speed, wind direction, and BLH, for the four seasons during the period of July 2011 to May 2013 at the PKUERS site. "OBS" and "SIM" represent the mean values of the observations and simulations, respectively.**

|  |  | DJF | MAM | JJA | SON |
|---|---|---|---|---|---|
| T (K) | OBS | 270.96 | 286.42 | 300.46 | 289.38 |
|  | SIM | 266.70 | 281.74 | 296.83 | 285.55 |
|  | NMB (%) | -1.57 | -1.63 | -1.21 | -1.33 |
|  | RMSE | 4.63 | 5.06 | 4.04 | 4.28 |
| RH (%) | OBS | 32.57 | 34.00 | 61.91 | 46.15 |
|  | SIM | 45.32 | 38.92 | 63.78 | 49.44 |
|  | NMB (%) | 39.15 | 14.47 | 3.01 | 7.12 |
|  | RMSE | 17.33 | 13.36 | 10.67 | 15.64 |
| Wind Speed (m s$^{-1}$) | OBS | 1.53 | 2.23 | 1.71 | 1.82 |
|  | SIM | 4.23 | 4.90 | 3.47 | 4.57 |
|  | NMB (%) | 177.27 | 119.34 | 102.84 | 150.82 |
|  | RMSE | 3.40 | 3.50 | 2.34 | 3.50 |
| Wind Direction (°) | OBS | 322.63 | 291.49 | 231.82 | 304.83 |
|  | SIM | 175.62 | 147.12 | 336.22 | 182.09 |
|  | NMB (%) | -14.53 | -1.23 | -1.89 | -8.55 |
|  | RMSE | 126.44 | 128.69 | 122.92 | 125.30 |
| BLH: 2 PM (m) | OBS | n.a. | n.a. | 1338.74 | n.a. |
|  | SIM | n.a. | n.a. | 1788.73 | n.a. |
|  | NMB (%) | n.a. | n.a. | 33.61 | n.a. |
|  | RMSE | n.a. | n.a. | 647.00 | n.a. |
| BLH: 8 AM (m) | OBS | 389.60 | 468.18 | 373.28 | 356.30 |
|  | SIM | 203.95 | 518.11 | 518.79 | 252.14 |
|  | NMB (%) | -47.65 | 10.66 | 38.98 | -29.23 |
|  | RMSE | 497.37 | 680.43 | 396.69 | 487.38 |
| BLH: 8 PM (m) | OBS | 436.39 | 618.33 | 502.45 | 417.24 |
|  | SIM | 482.20 | 1003.04 | 501.51 | 636.58 |
|  | NMB (%) | 10.50 | 62.22 | -0.19 | 52.57 |
|  | RMSE | 703.30 | 1159.34 | 840.49 | 940.83 |



**Table 2. Details of the sensitivity simulations from Cases 1 to 9.**

| Case No. | Tested Factors | Modifications in the model | Reference |
|---|---|---|---|
| 1 | BLH | Multiply by 3.6 for nighttime if the BLH is lower than 500 m | This study |
| 2 | $SO_2$ | Summer: multiply $SO_2$ emission by 0.8<br>Winter: multiply $SO_2$ emission by 1.5 | Koukouli et al. (2018) |
| 3 | $NH_3$ | Multiply non-agriculture $NH_3$ emission by 1.4 | Kang et al. (2016) |
| 4 | OH level | Summer: multiply $PM_{2.5}$-related reaction rates by 1.5<br>Winter: multiply $PM_{2.5}$-related reaction rates by 2 | This study |
| 5 | $\gamma_{SO2-RH}$ | Add-in: between $2\times10^{-5}$ to $5\times10^{-5}$ depending on RH | B. Zheng et al. (2015) |
| 6 | $\gamma_{SO2-ALWC}$ | Add-in: between $10^{-6}$ to $10^{-4}$ depending on ALWC | J. Li et al., (2018) |
| 7 | $\gamma_{NO2}$ | Change $\gamma_{NO2}$ from $10^{-4}$ to $10^{-6}$ | M. Li et al., (2019) |
| 8 | $\gamma_{N2O5}$ | Change $\gamma_{N2O5}$ from 0.02 (global mean) to $10^{-3}$ | McDuffie et al., (2018) |
| 9 | Wet deposition | Use the seasonal varied in-cloud condensation water and update the empirical washout rate for $HNO_3$ | Luo et al., (2019) |