# Peer review of "Model bias in simulating major chemical components of PM2.5 in China"

_Atmospheric Chemistry and Physics, 2020_

## Referee Comment (RC1) · Anonymous Referee #1 · 1 Jun 2020

Review of Miao et al, 2020

This paper addresses the important issue of the difficulty models have in correctly simulating $PM_{2.5}$ composition. This is to my knowledge the most comprehensive evaluation of a CTM's performance in China and provides a valuable starting place for the future investigation of many issues, such as the overestimation of nitrate and the underestimation of $NH_3$, CO, OH and $HO_2$. This paper provides an analysis of the performance of the model in all four seasons, which is rare. This paper is well-written and within the scope of ACP and should be published after the minor revisions listed below.

**Major Comments.**

My main major comment is that the authors should be more careful in stating potential reasons for model biases, and either perform "back-of-the-envelope" calculations, or quick sensitivity tests to support their conclusions. They have already gone to huge effort to perform a large set of sensitivities, but a little more context on the environment in China could be extremely helpful in interpreting the results. It would be particularly good for example to understand when SIA is sensitivity to NH3, or HNO3, or base. Finally, the conclusions could more clearly state the main findings from this work, with the key numbers that highlight their findings, such as improvement from reduced NO2 uptake and the remaining summertime nitrate bias. The minor comments below state additional specific suggestions for this.

**Minor Comments.**
1. Page 4, line 108. I don't quite understand why you need to divide the observation data by 0.8 to compare to the model. Should the model represent both PM1 and PM2.5, and could you not compare to both?
2. Page 4, line 126 – provide the model doi.
3. Page 5, line 161 – Is there a citation for "the top-down estimates."?
4. Page 5, line 171 – It would be very useful to have a table of relevant emissions totals for comparison by future studies.
5. Page 5, line 189 – What season was evaluated in Fu et al., 2012 and Zhao et al., 2016? Does the conclusion still hold about the improved model performance with the simple scheme if compared by season?
6. Page 5, line 191 – It might be helpful to readers to start a new paragraph discussing the seasonality of the model bias.
7. Page 7, line 200 – Are you saying that there is too much NH4 in YRD because there is too much NOx making too much NH4NO3? I
8. Page 7, line 204 – The lack of model gradient in SO4 between urban and rural sites is striking. Do you have an explanation for this? Is there an urban/rural gradient in SO2 in the model?
9. Page 7, line 207 – Do you expect model resolution to have an effect on the ability to simulate urban aerosol?
10. Page 7, line 208 – Are you saying that winter = haze? It does not appear that you have classified winter data as 'haze'/'not haze', please clarify.

11. Page 7, line 218 – Can you be more specific about the possible causes of the seasonality in the OA bias? Could the seasonality imply an issue with biogenic vs. anthropogenic SOA?

12. Page 7, line 220 – Is it really necessary to show the model biases on a log scale?

13. Figure S3 – what are the red dashed lines?

14. Page 7, line 222 - Is the summer value for PM2.5 really within 30%?

15. Page 8, line 227 – Can you please explain the reasoning for excluding data over 150 ug $m^{-3}$?

16. Page 8, line 231 - Instead of "insignificant", could you state the model bias?

17. Page 8, line 233 – Could you give us more statistics on the diurnal cycle, it is hard to see in these plots that nitrate and ammonium are "flatter" than sulfate.

18. Page 8, line 243 – could the ratio of nitrate / nitrate + hno3 tell you whether there is an issue with model partitioning at this site?

19. Figure 2 – Why is there a morning peak in wintertime OA?

20. Figure S8 – What are the red numbers? Can you explain the large difference in the median vs. mean difference particularly in winter and fall?

21. Table 1 – The NMB values for wind direction don't make sense, it seems like they should be much larger.

22. Page 9, line 274 – Do you have an explanation for the seasonality in SO2 that the model is missing? Could this be for example from heating sources that the inventory doesn't capture?

23. Figure 4b – Why compare against NO2 and not NOx? The modeling partitioning could also have issues.

24. Page 9, line 279 – Could you run a quick sensitivity test to determine whether say turning off NO2 uptake brings NO2 into better agreement?

25. Page 9, line 285 – Can you calculate whether aerosol is generally sensitivity to NH3 or HNO3 in each season?

26. Page 9, line 287 – You mean in the simple scheme, right? CO wouldn't affect the semi-volatile scheme if I understand correctly?

27. Figure 5 – The uncertainties on these observations, particularly the radicals, are large. Could you put error bars, or shading etc., on the observations?

28. Page 10, line 305 – Not necessarily, depending on conditions, inclusion of clno2 can increase nitrate due to clno2 photolysis – See Sarwar et al., 2014 (GRL).

29. Page 10, line 308 – Jaegle et al., 2018 discusses the Eastern United states in winter, I am confused by this reference here. I also can't find the seasonality you reference in these citations, please clarify.

30. Page 10, line 313 – However the lack of daytime HONO is a model issue – does this provide support for photolysis of nitrate?

31. Page 10, line 318 – Just to clarify, you are aiming to address general model biases, not biases specific to haze?

32. Page 11, line 340 – Did you run all 50 sensitivity simulations at nested resolution? Did you also consider the effect of resolution itself? See Zakoura and Pandis, 2018 (Atmospheric Environment)

33. Page 11, line 341 – Why not test whether scaling up winter and fall CO improves model SOA in the simple scheme?

34. Figure 6 – It might help to have a horizontal line through the 1-1 line so we can see when the model is over or under estimating.

35. Page 11, line 350 – Would the model bias in nitrate impact aerosol water and thus result in overestimated sulfate particularly using the ALWC parameterization?

36. Page 11, line 354 – Isn't this conclusion supported by the extreme model overestimate of HONO in Figure S9?

37.  Figure 6, case 8, Why is the median so much more impacted than the mean?

38. Page 12, line 360 – Why not test these things?

39. Page 12, line 362 – Why does increasing sulfate reduce nitrate in the model?

40. Figure 7.  I understand the authors aim in Figure 7,  but it is difficult to follow.  Possibly a table would be easier for the reader to understand.

41. General comment – is there any reason to think that in-cloud oxidation of SO2 is underestimated? Could model cloud biases be part of the issue?

42. Conclusions – it would help the reader to be more specific in the conclusions about the impact of your sensitivities.  For example, accurate SO2 emissions result in XX improvement in the model agreement with SO4. Generally, if the authors could put in the conclusions more numbers on their findings, for example, even our most improved model is still biased by XX % in summer, it could help improve citations by future modeling studies.

43. Page 13, line 410 – Can you provide the explanation for this here? Why is this the case?

---

## Referee Comment (RC2) · Anonymous Referee #2 · 6 Jul 2020

Model bias in simulating major chemical components of PM2.5 in China

Ruqian Miao, Qi Chen, Yan Zheng , Xi Cheng , Yele Sun, Paul I. Palmer, Manish Shrivastava4, Jianping Guo, Qiang Zhang, Yuhan Liu, Zhaofeng Tan, Xuefei Ma, Shiyi Chen, Limin Zeng, Keding Lu, Yuanhang Zhang

The paper presents a comprehensive investigation into the uncertainties in PM2.5 simulated with the GEOS-Chem model for China and potential sources of errors. PM is a complex pollutant and even after the decades of its modelling, most of air quality/chemical transport models are still often struggling with accurate representation of PM, in particular during pollution episodes. Given large uncertainties in descriptions of aerosol chemical and physical processes, the availability of good quality observations is crucial for models' evaluation and constraining. In the presented work, the authors

compiled and used for the model evaluation an impressive volume of observational data of non-refractive submicron PM components (sulphate, nitrate, ammonium and organic aerosols), as well as aerosol gaseous precursors in China. They also performed a series of sensitivity tests, modifying multiple parameters, to obtain the best model correspondence with the observations.

The paper is in general fairly well written (though the bounty of technicalities sometimes makes reading somewhat heavy); the figures and tables are quite helpful in visualizing and presenting the results. The topic of the paper is highly relevant, the scientific material and findings are quite interesting, and thus after some minor revisions it can be recommended for publication in ACP.

My first reaction is that given the impressive amount of testing, the conclusions appear somewhat little constructive and of a rather general character. In other words, it is unclear what would be the first priorities the authors plan to improve the GEOS_Chem's performance with respect to PM2.5. Would the authors comment on that?

The 'best combination' of all tested parameters did not yield a satisfactory model agreement with short term observations in Beijing. Has it been tested against the whole 2006-2016 campaign dataset?

In particular, the GEOS-chem is shown to have troubles to reproduce observed concentrations of SO4 and NO3. Is that for China simulations only? Or was that seen also so for other world's regions? I'd suggest to include in Introduction a small paragraph about that if such evaluations are available.

Have the authors seen a paper by H. Bian et al.: Investigation of global particulate nitrate from the AeroCom phase III experiment (Atmos. Chem. Phys., 17, 12911–12940, 2017 https://doi.org/10.5194/acp-17-12911-2017)? It does not appear that overestimating NO3 in Asia is a generic feature among the nine CTM and climate models participating in the paper.

[Figure]

There are other processes, not been investigated in the paper, which could be sources of e.g. NO3 overestimation, for instance the equilibrium formation of ammonium nitrate. How well does ISORROPIA work for China chemical regime? Has this been studied before? Any reference to the results?

Another source of uncertainties is dry deposition velocities of NO3 and NH4, which were measured to be higher than typically predicted by the models (E. Nemitz et al.: Concentrations and surface exchange fluxes of particles over heathland; Atmos. Chem. Phys., 4, 1007–1024, 2004 www.atmos-chem-phys.org/acp/4/1007/). Would the authors consider to investigate into this process?

A minor general comment: winter haze events are mentioned every now and then, without any clear context. Could the authors explain early in the paper why haze occurrence is an issue in the manuscript?

Other comments and suggestions to the text:

Line 27. suggesting existing inaccuracies in the processes description (or presentation).

Lines 30-31 and 172-174. It's unclear what heterogeneous SO2 oxidation reactions are in the model (by H2O2, ozone?)

Lines 34-35. Again, can the author show that ISORROPIA is working properly? Clarify 'related to removal'. Wet or dry, or both.

Line 42. what is considered to be reasonable?

Line 45. 'the model performance on PM2.5 is component-dependent' sounds strange. Maybe like: even though the model represents well observed PM2.5, it may happen due to compensation errors in model simulated PM2.5 components.

Line 48. I agree to some extend about SOA, but not about sulphate and nitrate (see my Ref. to Bian above). Perhaps the authors mean only specific studies for China.

[Figure]

Line 57. Suggestion: The uncertainties in the emissions of primary PM and gaseous precursors of secondary PM are quite large. . ..

Line 82. Suggested: therefore it is important to evaluate the model for all individual components of PM2.5

Line 83. The measurements' artefacts can also decrease the discrepancies. . . the point is that in such cases, the model evaluation results give a wrong message.

Line 103. Write: Institute of Atmospheric Physics (IAP) – for future use of abbreviation. Could you write here what site type is this (urban/suburban background?)

Line 130. Suggestion: For comparison with observations at IAP. Beijing, the model simulations were performed for the ASCM measurements period. . ...

Line 132: Do I understand right that model simulations for 2012 meteorological conditions were used for comparison with 2006-2016 observations. Could the authors then say how (un)typical the 2012 weather was. Were year dependent emissions used, or also the same for 2012?

Line 150. Suggested: 80% which is considered to be a reasonable assumption (instead of 'for simplicity')

Line 151. However, Hodzic. . . showed that the results were not very sensitive to. . ...

Line 161-162. What is the relative contribution of non-agricultural NH3 emissions compared to the agricultural ones? This is important to know when analysing ammonium nitrate formation in cities.

Lines 172-180. Move up to Chemistry description, above the emissions.

Line 182. Suggestion: Model performance for the individual PM components

Line 188: The modelled ammonium concentrations compare with observations better than simulated sulfate. . ..

Line 191. Further, we find that the model biases...

Lines 192 and 208-210 some repetition.

Line 196. Would you expect the model performance to differ over China? Why? How different those regions are (weather, emissions?). Does that mean that the model is insensitive to the differences in meteo and chemical regimes?

Line 208. Suggestion: On a seasonal basis,..

Line 210-11. ..the overestimation of nitrate concentration is largest in spring, summer and autumn.., while the model bias is much smaller in winter...

Line 212-213. Are those simulations also with GEOS_Chem and ISORROPIA?

Line 214. In all seasons?

Line 217. The model performs worst in autumn

Line 218. This is about the only time when the correlation is mentioned. Why it's considered important here, but not for the other components? Is the relative importance of ASOA greater in summer?

Line 220. –compared to the 2 years of hourly observations....

Line 223. ..underestimation of sulfate and OA by the overestimation of nitrate...

Line 226-27. Explain exclusion of the observations over 150 ug-m3

Line 242. Does that mean: If the evaporation of ammonium nitrate ......was accounted for in the model, the day time variation.. could be flatter?

Line 254. Semivolatile POA scheme previously used? in GEOS-Chem

Line 267. ...simulations lead

Line 271. Suggestion: The uncertainties related with emission data 8including their temporal profiles) are considered to be one of the major sources of inaccuracies in

modelled concentrations. . . ..

Line 275. It is widely shown that regional models cannot accurately reproduce NO2 at urban sites. Would the authors really expect the model with a resolution of 50-60 km to be capable of managing that?

Lines 282-285. What is the main sources and relative importance of non-agricultural NH3? Why is it especially important during haze events?

Lines 288-290: Unclear what is said here.

Lines 311-12. Should be formulated more clear: The photolysis rate of particle-phase HNO3 was shown in aged air masses to be higher than for the gaseous HNO3. . ...., but in Beijing particulate NO3 may have lower photolysis rates, because. . ...

Line 315. From the factors

Line 319-20. . . .weeks were free of severe haze episodes (with extreme conditions which the model fails to reproduce????)

Line 338. The increased wet deposition if nitrate. . ..

Line 360. Faster photolysis of particulate nitrate? Sounds contradictory to what is written on lines 311-312

---

## Author Comment (AC1) · 18 Aug 2020

**Response to reviews**

Reviewer comments are in **bold**. Author responses are in plain text labeled with [R]. Line numbers in the responses correspond to those in the revised manuscript (the version with all changes accepted). Modifications to the manuscript are in *italics*.

**Reviewer #1**

**This paper addresses the important issue of the difficulty models have in correctly simulating PM$_{2.5}$ composition. This is to my knowledge the most comprehensive evaluation of a CTM's performance in China and provides a valuable starting place for the future investigation of many issues, such as the overestimation of nitrate and the underestimation of NH$_3$, CO, OH, and HO$_2$. This paper provides an analysis of the performance of the model in all four seasons, which is rare. This paper is well-written and within the scope of ACP and should be published after the minor revisions listed below.**

[R0] We thank the reviewer for the valuable feedback and constructive suggestions. Detailed responses are given below.

**Major Comments.**
**My main major comment is that the authors should be more careful in stating potential reasons for model biases, and either perform "back-of-the-envelope" calculations, or quick sensitivity tests to support their conclusions. They have already gone to huge effort to perform a large set of sensitivities, but a little more context on the environment in China could be extremely helpful in interpreting the results. It would be particularly good for example to understand when SIA is sensitivity to NH$_3$, or HNO$_3$, or base.**

[R1] We thank the reviewer for the suggestion and agree that the sensitivity of SIA to NH$_3$ or HNO$_3$ is an important issue for the simulation of PM$_{2.5}$. Nenes et al. (2020a; 2020b) indicate that aerosol pH and ALWC determine the sensitivity of PM$_{2.5}$ to NH$_3$ or HNO$_3$ and the reactive nitrogen deposition. Their analysis show that in China, especially in northern China, PM$_{2.5}$ is more likely sensitive to HNO$_3$ and in some cases to HNO$_3$+NH$_3$. The model overestimates the HNO$_3$ and nitrate concentrations largely in Beijing, suggesting that the model over-predicts the nitrate availability. Analysis of the potential factors to the model bias shows this overprediction cannot be explained by the chemical production and meteorology. We therefore state that the possible explanations include inaccurate dry deposition of HNO$_3$ and nitrate and the photolysis of particulate nitrate in the model. In particular, the latter remains largely unknown under ambient conditions. However, it is difficult to diagnose more within the scope of this manuscript. We have added related discussions and clarified some details in the revised manuscript (e.g., in Line 249-252, 375-384, and 435-444).

**Finally, the conclusions could more clearly state the main findings from this work,**

with the key numbers that highlight their findings, such as improvement from reduced NO₂ uptake and the remaining summertime nitrate bias.

[R2] We have revised the conclusion paragraph with key numbers to highlight our findings.

**The minor comments below state additional specific suggestions for this.**
**Minor Comments.**
**1. Page 4, line 108. I don't quite understand why you need to divide the observation data by 0.8 to compare to the model. Should the model represent both PM₁ and PM₂.₅, and could you not compare to both?**

[R3] The modeled sulfate, nitrate, ammonium, and OA are not specified to any size domain. In polluted environments, substantial mass of these species present in the super-micron domain. The modeled concentrations represent $PM_{2.5}$ not $PM_1$. The previous AMS or ACSM measurements are however for $PM_1$. Recent measurements in North China Plain suggest that 0.8 is a good coefficient to use for converting $PM_1$ to the $PM_{2.5}$ mass for these species. We therefore applied 0.8 when comparing the observations to the model results. This coefficient doesn't affect the model evaluation much given the measurement uncertainty of 30% is considered. For clarification, we have revised Line 114-116 as follows "*Model results plausibly represent fine particles not submicron portion in polluted environments. The submicron-to-fine mass ratios are about 0.8 for sulfate, nitrate, ammonium, and OA in summer and winter in NCP and may decrease to 0.5 during the severe winter-haze episodes under high RH (Fig. S1 in SI) (Zheng et al., 2020). We therefore divided the submicron observation data by 0.8 for the four species when comparing to the model results*".

**2. Page 4, line 126 – provide the model doi.**
[R4] The model doi is added in Line 133.

**3. Page 5, line 161 – Is there a citation for "the top-down estimates."?**
[R5] We added the citation for the top-down estimates in Line 175.

**4. Page 5, line 171 – It would be very useful to have a table of relevant emissions totals for comparison by future studies.**
[R6] We have added Table S2 in SI to show the total emissions of primary $PM_{2.5}$ and the gaseous precursors.

**5. Page 5, line 189 – What season was evaluated in Fu et al., 2012 and Zhao et al., 2016? Does the conclusion still hold about the improved model performance with the simple scheme if compared by season?**
[R7] Fu et al. (2012) and Zhao et al. (2016) evaluated four seasons. The improved model performance with the Simple SOA scheme holds for all the four seasons. We have revised Line xxx as follows "*The Simple SOA scheme shows improved performance on OA for all seasons (NMB = -0.26, R = 0.70)*".

**6. Page 5, line 191 – It might be helpful to readers to start a new paragraph discussing the seasonality of the model bias.**

[R8] We have started a new paragraph and moved the original paragraph (two paragraphs later) about seasonality here.

**7. Page 7, line 200 – Are you saying that there is too much NH₄ in YRD because there is too much NOx making too much NH₄NO₃?**

[R9] We think that the $NH_3$-rich environment in YRD promote the formation of ammonium nitrate and ammonium sulfate. We have revised the sentence in Line 221-223 for clarification.

**8. Page 7, line 204 – The lack of model gradient in SO₄ between urban and rural sites is striking. Do you have an explanation for this? Is there an urban/rural gradient in SO₂ in the model?**

[R10] The non-urban sites herein contain rural and suburban sites. The modeled $SO_2$ concentrations do not show significant urban/non-urban gradients (9.84/10.13 ppbv), possibly because many $SO_2$ sources like power plants and industry are located outside of urban areas. We think the greater sulfate concentrations observed in urban sites than in non-urban sites are perhaps a result of chemistry that occurs during the transport process but has not been well presented in the model (e.g., the heterogeneous formation of sulfate).

**9. Page 7, line 207 – Do you expect model resolution to have an effect on the ability to simulate urban aerosol?**

[R11] We expect that the model resolution affects sites nearby sources. The urban sites herein are generally urban background sites, meaning that urban air are well mixed at the sites. The model-observation comparisons are also based on campaign-average values for which pollution plumes should be smoothed out. Moreover, the model grid of 0.5°×0.625° isn't too big compared to the size of the cities in China. The sites are well covered by the model grid boxes. In addition, the differences between the simulated concentrations of sulfate, nitrate, ammonium, and OA by 0.5°×0.625° and 0.25°×0.3125° horizontal resolutions for the sites in Beijing are within 35% with high $R$ value (> 0.9). We therefore do not expect a significant influence of the model resolution on the analysis herein.

**10. Page 7, line 208 – Are you saying that winter = haze? It does not appear that you have classified winter data as 'haze'/'not haze', please clarify.**

[R12] Severe haze occurred more often in winter than in other seasons. We have clarified in the text that *"the underestimation of sulfate occurs all year round, and the greatest underestimation occurs in winter (NMB = -0.54) (Fig. 1a and Table S3). The seasonality of the model bias is partially explained by the underestimation of SO2 emissions in winter (Wang et al., 2014; Koukouli et al., 2018). Similar to other models, our model failed to reproduce the high sulfate concentrations during the haze periods because of the underrepresented heterogeneous production (Wang et al., 2014; G. J.*

*Zheng et al., 2015). Severe haze events occurred more often in winter in China, contributing to the seasonality of the model bias".*

**11. Page 7, line 218 – Can you be more specific about the possible causes of the seasonality in the OA bias? Could the seasonality imply an issue with biogenic vs. anthropogenic SOA?**

[R13] The seasonality in the OA bias is perhaps more contributed by anthropogenic SOA as it dominates the OA mass in most part of China. We have revised in Line 211-214 as follows: "*the underestimation of OA occurs all year round, but the worst bias happens in autumn. Biases in the precursor emissions as well as the assumed nonseasonal conversion rate from precursors to particle-phase SOA are the possible reasons of the seasonality of the OA bias*".

**12. Page 7, line 220 – Is it really necessary to show the model biases on a log scale?**

[R14] We thank the reviewer for the suggestion and have modified the Fig. S3 and S4 with a linear scale.

**13. Figure S3 – what are the red dashed lines?**

[R15] The red dashed lines denote the ratios of simulation and observation of 1.3, 1, and 0.7. We have added the description in the figure caption.

**14. Page 7, line 222 - Is the summer value for PM$_{2.5}$ really within 30%?**

[R16] The median value of the simulation-to-observation ratios for summertime PM$_{2.5}$ is within 30%. In Fig. S3, the upper and lower red dashed lines show the 30%.

**15. Page 8, line 227 – Can you please explain the reasoning for excluding data over 150 ug m$^{-3}$?**

[R17] As described in [R3], the model results plausibly represent fine particles not submicron portion in polluted environments in China. The submicron-to-fine mass ratios are about 0.8 for sulfate, nitrate, ammonium, and OA in summer and winter in NCP and may decrease to 0.5 during the severe haze episodes under high RH. We divided the submicron observation data by 0.8 for the four species when comparing to the model results. The test here by excluding data over 150 µg m$^{-3}$ is to exclude the severe haze periods, and thus to prove that the discussion won't be affected by the submicron-to-fine ratio. For clarification, we have revised the text in Line 236-239 as follows: "*Figure S4 in SI shows the simulation-to-observation ratios when excluding the periods of NR-PM$_{2.5}$ mass concentrations over 150 µg m$^{-3}$. During these periods, the submicron-to-fine ratios may decrease from 0.8 (used herein) to 0.5. The model biases and their seasonal variations in Fig. S4 are similar to the previous results, suggesting insignificant impacts of haze periods on the statistic evaluations*".

**16. Page 8, line 231 - Instead of "insignificant", could you state the model bias? and 17. Page 8, line 233 – Could you give us more statistics on the diurnal cycle, it is hard to see in these plots that nitrate and ammonium are "flatter" than sulfate.**

[R18-19] We have added values and revised the statements in Line 241-249 as follows: "*the observed sulfate shows a daytime concentration build-up (2-4 μg m⁻³) in spring and summer, suggesting a photochemical production (Sun et al., 2015). The wintertime diurnal pattern shows a steady but later enhancement (~5 μg m⁻³) in the afternoon. The simulated profiles show less daytime concentration elevations (0-2 μg m⁻³), suggesting insufficient production, overestimated boundary-layer dilution, or removal during the day in the model (Fig. 2a). By contrast, the observed diurnal variations of hourly-mean nitrate and ammonium concentrations are less than sulfate (Fig. 2b-c). The 2-5 times greater concentrations of simulated nitrate at night suggest over-predicted nighttime production, underestimated boundary-layer dilution, or underestimated removal of nitrate. Nighttime production of nitrate by the heterogeneous uptake of $N_2O_5$ and $NO_2$ is an important pathway of nitrate production in northern China (Wang et al., 2018; Alexander et al., 2020).* ".

**18. Page 8, line 243 – could the ratio of nitrate / nitrate + HNO₃ tell you whether there is an issue with model partitioning at this site?**

[R20] The nitrate partitioning fraction is determined by the thermodynamic equilibrium, which depends on aerosol acidity and ALWC. Because the model overestimates sulfate concentrations in summer, underestimates sulfate in winter, and underestimates ammonia concentrations in all seasons. The nitrate partitioning fraction can be biased by various reasons. We therefore did not use it to indicate the partitioning problem. We have revised the text to clarify the overestimation of total nitrate in the model in Line 251-252.

**19. Figure 2 – Why is there a morning peak in wintertime OA?**

[R21] The morning peak of wintertime OA is mainly caused by the POA emissions from the residential sector (e.g., from residential coal burning). The diurnal profile of the emissions from the residential sector is shown in Fig. S2. We have added this discussion in Line 266-267.

**20. Figure S8 – What are the red numbers? Can you explain the large difference in the median vs. mean difference particularly in winter and fall?**

[R22] The red numbers are the median values of MERRA2- and radiosonde-derived BLH. We have added the description in the figure caption. MERRA2-derived BLH are sometimes extremely large, leading to high MERRA2-to-observation ratios and subsequently large differences in the median vs. mean.

**21. Table 1 – The NMB values for wind direction don't make sense, it seems like they should be much larger.**

[R23] We have added the MB values to Table 1. Using the MB values for wind direction make more sense than using NMB.

**22. Page 9, line 274 – Do you have an explanation for the seasonality in SO₂ that the model is missing? Could this be for example from heating sources that the**

**inventory doesn't capture?**

[R24] The seasonality of $SO_2$ emission mainly results from the seasonality of emission from the residential sector. Therefore, we agree that the failure of inventory to capture some residential and commercial heating sources can contribute to the weak seasonality in MEIC.

**23. Figure 4b – Why compare against $NO_2$ and not $NO_x$? The modeling partitioning could also have issues.**

[R25] The relative coarse model resolution (about 50 km) limits the model performance of NO in the source region due to its quick conversion to $NO_2$ before transport through the gird. Compared to NO, $NO_2$ has a longer lifetime about several hours to nearly one day (Shah et al., 2020), which let it more even with the grid and be better presented by model. Therefore, we used $NO_2$ rather than $NO_x$ in the model evaluation. Actually, using $NO_x$ will not affect the conclusion.

**24. Page 9, line 279 – Could you run a quick sensitivity test to determine whether say turning off $NO_2$ uptake brings $NO_2$ into better agreement?**

[R26] We have run a sensitivity test for turning off $NO_2$ uptake. The result shows that the concentration of $NO_2$ increases by 70.3% and 58.1% in winter and summer, respectively, which agrees better with the observations.

**25. Page 9, line 285 – Can you calculate whether aerosol is generally sensitivity to $NH_3$ or $HNO_3$ in each season?**

[R27] Previous studies show that nitrate is generally sensitive to $HNO_3$ in summer (Wen et al., 2018) and sensitive to $NH_3$ during winter haze periods (Xu et al., 2019). Nenes et al. (2020b) also show that the nitrate formation is sensitive to $HNO_3$ and in some cases to $HNO_3+NH_3$.

**26. Page 9, line 287 – You mean in the simple scheme, right? CO wouldn't affect the semivolatile scheme if I understand correctly?**

[R28] Yes, we discuss the SOA precursors here based on the Simple SOA scheme. CO may have little impact on the semivolatile scheme by its effect on OH radicals.

**27. Figure 5 – The uncertainties on these observations, particularly the radicals, are large. Could you put error bars, or shading etc., on the observations?**

[R29] We thank the reviewer for the suggestion and have added the shading to present the standard deviation of these observations and simulations in Fig. 5.

**28. Page 10, line 305 – Not necessarily, depending on conditions, inclusion of $ClNO_2$ can increase nitrate due to $ClNO_2$ photolysis – See Sarwar et al., 2014 (GRL).**

[R30] We agree with the reviewer that the effect of inclusion of $ClNO_2$ on nitrate concentration is depending on conditions. Figure 2 in Sarwar et al. (2014) suggests that the heterogeneous $ClNO_2$ production can decrease both summertime and wintertime nitrate concentration in northern China. We have revised the statement in Line xxx as

follows: "*The model uses relatively high values of $\gamma_{N2O5}$, which can lead to the overestimation of nitrate (McDuffie et al., 2018; Davis et al., 2008; Jaegle et al., 2018). Besides, the missing formation of nitryl chloride from the $N_2O_5$ uptake trends to contribute to the nitrate overestimation in northern China (Sarwar et al., 2014)*".

**29. Page 10, line 308 – Jaegle et al., 2018 discusses the Eastern United states in winter, I am confused by this reference here. I also can't find the seasonality you reference in these citations, please clarify.**
[R31] Jaegle et al., 2018 is the citation for supporting the previous sentence "Biases may also relate to the atmospheric removal of the SIA species". We have corrected it in Line 204-205.

**30. Page 10, line 313 – However the lack of daytime HONO is a model issue – does this provide support for photolysis of nitrate?**
[R32] Ye et al. 2017 showed low photolysis rates of particle-phase nitrate when there is thick organic coating on $PM_{2.5}$. The lack of daytime HONO is perhaps a result of underrepresented sources (e.g., the heterogeneous formation on the surface of land, buildings and so on).

**31. Page 10, line 318 – Just to clarify, you are aiming to address general model biases, not biases specific to haze?**
[R33] Yes, we choose the periods during which the mean $PM_{2.5}$ concentrations were lower than 75 ug $m^{-3}$. We have revised this sentence to clarify.

**32. Page 11, line 340 – Did you run all 50 sensitivity simulations at nested resolution? Did you also consider the effect of resolution itself? See Zakoura and Pandis, 2018 (Atmospheric Environment)**
[R34] Yes, we run all simulations at nested resolution. The differences between the simulated concentrations of sulfate, nitrate, ammonium, and OA by 0.5°×0.625° and 0.25°×0.3125° horizontal resolutions for the sites in Beijing are within 35% with high $R$ value (>0.9) for all seasons. We therefore do not expect a significant influence of the model resolution on the analysis herein.

**33. Page 11, line 341 – Why not test whether scaling up winter and fall CO improves model SOA in the simple scheme?**
[R35] We did tested this. However, scaling up the emissions of CO in the Simple SOA scheme to increase anthropogenic SOA leads to the significant overestimation of SOA in non-urban areas. We have added this point in the text.

**34. Figure 6 – It might help to have a horizontal line through the 1-1 line so we can see when the model is over or under estimating.**
[R36] We thank the reviewer for the suggestion and have added the 1-1 line in Fig. 6.

**35. Page 11, line 350 – Would the model bias in nitrate impact aerosol water and**

thus result in overestimated sulfate particularly using the ALWC parameterization?

[R38] Yes, the overestimated ALWC in summer due to the overestimation of nitrate concentration may affect the heterogeneous sulfate formation and lead to the overestimated summertime sulfate concentration in Case 6 (based on ALWC parameterization). We have added this discussion in Line 367-368.

**36. Page 11, line 354 – Isn't this conclusion supported by the extreme model overestimate of HONO in Figure S9?**

[R39] Yes, the updated $\gamma_{NO_2}$ can significantly reduce the overestimation of nighttime HONO concentration. We have revised this sentence and added this information.

**37. Figure 6, case 8, Why is the median so much more impacted than the mean?**

[R40] In panel c, case 8 fails to capture low nitrate concentrations of $< 0.5$ ug m$^{-3}$, leading to large simulation-to-observation ratios ($>5$). This significantly affects the mean value of the simulation-to-observation ratios other than the median value.

**38. Page 12, line 360 – Why not test these things?**

[R41] There are lack of parameter constraints for testing the dry deposition of HNO$_3$ and the photolysis of particulate nitrate. We therefore did not test them in this study.

**39. Page 12, line 362 – Why does increasing sulfate reduce nitrate in the model?**

[R42] The heterogeneous formation of sulfate may affect aerosol pH and ALWC that determine the sensitivity of nitrate formation to ammonia and nitrate availability, especially in winter when NH$_3$ emissions are low (Nenes et al., 2020b). We have revised this discussion in Line 387-388.

**40. Figure 7. I understand the authors aim in Figure 7, but it is difficult to follow. Possibly a table would be easier for the reader to understand.**

[R43] We have updated Fig. 7 and the corresponding text in Line 394-420.

**41. General comment – is there any reason to think that in-cloud oxidation of SO$_2$ is underestimated? Could model cloud biases be part of the issue?**

[R44] The in-cloud oxidation of SO$_2$ mainly from the oxidation of H$_2$O$_2$ and O$_3$. The observed H$_2$O$_2$ concentration (0.51 ppbv) in Beijing (Wang et al., 2016) are consistent with the simulated concentration (0.43 ppbv). Fig. 6 shows that O$_3$ concentration is overestimated. Both do not indicate the underestimation of in-cloud sulfate formation. Also, because sulfate concentrations can be generally reproduced in the US by Geos-Chem (Heald et al., 2012), we think the model cloud biases less likely affect the simulations herein.

**42. Conclusions – it would help the reader to be more specific in the conclusions about the impact of your sensitivities. For example, accurate SO$_2$ emissions result in XX improvement in the model agreement with SO$_4$. Generally, if the authors**

**could put in the conclusions more numbers on their findings, for example, even our most improved model is still biased by XX % in summer, it could help improve citations by future modeling studies.**

[R45] We have revised the conclusion with specific information.

**43. Page 13, line 410 – Can you provide the explanation for this here? Why is this the case?**

[R46] The overestimation of sulfate with worse $R$ in summer suggests that the parameterization of heterogeneous sulfate formation on RH and ALWC are insufficient, and therefore mechanistic approaches might be needed to improve the seasonality of the sulfate simulations.

**References**

Alexander, B., Sherwen, T., Holmes, C. D., Fisher, J. A., Chen, Q., Evans, M. J., and Kasibhatla, P.: Global inorganic nitrate production mechanisms: comparison of a global model with nitrate isotope observations, Atmos. Chem. Phys., 20, 3859-3877, https://doi.org/10.5194/acp-20-3859-2020, 2020.

Heald, C. L., Collett, J. L., Lee, T., Benedict, K. B., Schwandner, F. M., Li, Y., Clarisse, L., Hurtmans, D. R., Van Damme, M., Clerbaux, C., Coheur, P.-F., Philip, S., Martin, R. V., and Pye, H. O. T.: Atmospheric ammonia and particulate inorganic nitrogen over the United States, Atmos. Chem. Phys., 12, 10295-10312, https://doi.org/10.5194/acp-12-10295-2012, 2012.

Nenes, A., Pandis, S. N., Kanakidou, M., Russell, A., Song, S., Vasilakos, P., and Weber, R. J.: Aerosol acidity and liquid water content regulate the dry deposition of inorganic reactive nitrogen, Atmos. Chem. Phys. Discuss., 2020, 1-25, https://doi.org/10.5194/acp-2020-266, 2020a.

Nenes, A., Pandis, S. N., Weber, R. J., and Russell, A.: Aerosol pH and liquid water content determine when particulate matter is sensitive to ammonia and nitrate availability, Atmos. Chem. Phys., 20, 3249-3258, https://doi.org/10.5194/acp-20-3249-2020, 2020b.

Sarwar, G., Simon, H., Xing, J., and Mathur, R.: Importance of tropospheric $ClNO_2$ chemistry across the Northern Hemisphere, Geophys. Res. Lett., 41, 4050-4058, https://doi.org/10.1002/2014gl059962, 2014.

Shah, V., Jacob, D. J., Li, K., Silvern, R. F., Zhai, S., Liu, M., Lin, J., and Zhang, Q.: Effect of changing $NO_x$ lifetime on the seasonality and long-term trends of satellite-observed tropospheric $NO_2$ columns over China, Atmos. Chem. Phys., 20, 1483-1495, https://doi.org/10.5194/acp-20-1483-2020, 2020.

Sun, Y. L., Wang, Z. F., Du, W., Zhang, Q., Wang, Q. Q., Fu, P. Q., Pan, X. L., Li, J., Jayne, J., and Worsnop, D. R.: Long-term real-time measurements of aerosol particle composition in Beijing, China: seasonal variations, meteorological effects, and source analysis, Atmos. Chem. Phys., 15, 10149-10165, https://doi.org/10.5194/acp-15-10149-2015, 2015.

Wang, H., Lu, K., Chen, X., Zhu, Q., Wu, Z., Wu, Y., and Sun, K.: Fast particulate nitrate formation via N2O5 uptake aloft in winter in Beijing, Atmos. Chem. Phys., 18, 10483-10495, https://doi.org/10.5194/acp-18-10483-2018, 2018.

Wang, Y., Chen, Z. M., Wu, Q. Q., Liang, H., Huang, L. B., Li, H., Lu, K. D., Wu, Y. S., Dong, H. B., Zeng, L. M., and Zhang, Y. H.: Observation of atmospheric peroxides during Wangdu Campaign 2014 at a rural site in the North China Plain, Atmos. Chem. Phys., 16, 10985-11000, https://doi.org/10.5194/acp-16-10985-2016, 2016.

Wen, L., Xue, L., Wang, X., Xu, C., Chen, T., Yang, L., Wang, T., Zhang, Q., and Wang, W.: Summertime fine particulate nitrate pollution in the North China Plain: increasing trends, formation mechanisms and implications for control policy, Atmos. Chem. Phys., 18, 11261-11275, https://doi.org/10.5194/acp-18-11261-2018, 2018.

Xu, Z., Liu, M., Zhang, M., Song, Y., Wang, S., Zhang, L., Xu, T., Wang, T., Yan, C., Zhou, T., Sun, Y., Pan, Y., Hu, M., Zheng, M., and Zhu, T.: High efficiency of livestock ammonia emission controls in

alleviating particulate nitrate during a severe winter haze episode in northern China, Atmos. Chem. Phys., 19, 5605-5613, https://doi.org/10.5194/acp-19-5605-2019, 2019.

---

## Author Comment (AC2) · 18 Aug 2020

**Response to reviews**

Reviewer comments are in **bold**. Author responses are in plain text labeled with [R]. Line numbers in the responses correspond to those in the revised manuscript (the version with all changes accepted). Modifications to the manuscript are in *italics*.

**Reviewer #2**

**The paper presents a comprehensive investigation into the uncertainties in PM2.5 simulated with the GEOS-Chem model for China and potential sources of errors. PM is a complex pollutant and even after the decades of its modelling, most of air quality/chemical transport models are still often struggling with accurate representation of PM, in particular during pollution episodes. Given large uncertainties in descriptions of aerosol chemical and physical processes, the availability of good quality observations is crucial for models' evaluation and constraining. In the presented work, the author compiled and used for the model evaluation an impressive volume of observational data of non-refractive submicron PM components (sulphate, nitrate, ammonium and organic aerosols), as well as aerosol gaseous precursors in China. They also performed a series of sensitivity tests, modifying multiple parameters, to obtain the best model correspondence with the observations.**
**The paper is in general fairly well written (though the bounty of technicalities sometimes makes reading somewhat heavy); the figures and tables are quite helpful in visualizing and presenting the results. The topic of the paper is highly relevant, the scientific material and findings are quite interesting, and thus after some minor revisions it can be recommended for publication in ACP.**

[R0] We thank the reviewer for the valuable feedback and constructive suggestions. Detailed responses are given below.

**My first reaction is that given the impressive amount of testing, the conclusions appear somewhat little constructive and of a rather general character. In other words, it is unclear what would be the first priorities the authors plan to improve the GEOS-Chem's performance with respect to $PM_{2.5}$. Would the authors comment on that?**
[R1] We thank the reviewer for the suggestion and have rewritten the conclusion. As stated in the revised version, heterogeneous formation of sulfate and nitrate as well as the anthropogenic S/IVOC-related SOA are the first priorities to improve the model performance. However, our best model with all the updated factors still bias the nitrate in summer by 210%, which merits further investigations.

**The 'best combination' of all tested parameters did not yield a satisfactory model agreement with short term observations in Beijing. Has it been tested against the whole 2006-2016 campaign dataset?**

[R2] We did not test the model performance of the "best combination" of all tested parameters against the whole 2006-2016 campaign dataset. The main reason is that we don't have sufficient measurements to constrain the various factors outside Beijing, for example, $SO_2$ emission and OH levels.

**In particular, the GEOS-Chem is shown to have troubles to reproduce observed concentrations of $SO_4$ and $NO_3$. Is that for China simulations only? Or was that seen also so for other world's regions? I'd suggest to include in Introduction a small paragraph about that if such evaluations are available.**

[R3] The overestimation of nitrate in GOES-Chem was also observed in the US, where the model reproduce sulfate concentrations (Heald et al., 2012). We have added this information in the Introduction in Line 53-54.

**Have the authors seen a paper by H. Bian et al.: Investigation of global particulate nitrate from the AeroCom phase III experiment (Atmos. Chem. Phys., 17, 12911–12940, 2017 https://doi.org/10.5194/acp-17-12911-2017)? It does not appear that overestimating $NO_3$ in Asia is a generic feature among the nine CTM and climate models participating in the paper.**

[R4] The EANET measurement sites used in Bian et al. (2017) are mostly located in areas having low $NO_x$ concentrations. By contrast, nearly a half of the measurements in this study are from polluted northern China where the $NO_x$ concentrations are high. The two regions can be different in chemical domains of the sensitivity of aerosol to $NH_3$ and $NO_x$ emissions and therefore be different in nitrate formation potential (Nenes et al., 2020). Another multi-model comparisons in Asia also show the overestimation of nitrate in Asia (Chen et al., 2019).

**There are other processes, not been investigated in the paper, which could be sources of e.g. $NO_3$ overestimation, for instance the equilibrium formation of ammonium nitrate. How well does ISORROPIA work for China chemical regime? Has this been studied before? Any reference to the results?**

[R5] Previous study in Beijing shows the ISORRPOPIA II model can reproduce the concentrations of sulfate, nitrate, ammonium, and $NH_3$ with $R$s > 0.9 and NMBs within 10% and generally capture the partitioning of $NH_3$/ammonium (Liu et al., 2017). We except minor bias from ISORRPOPIA compared with the potential biases in the heterogeneous uptake of $NO_2$ and $N_2O_5$ as well as other factors related to the precursor oxidation and the removal processes.

**Another source of uncertainties is dry deposition velocities of $NO_3$ and $NH_4$, which were measured to be higher than typically predicted by the models (E. Nemitz et al.: Concentrations and surface exchange fluxes of particles over heathland; Atmos. Chem. Phys., 4, 1007–1024, 2004 www.atmos-chem-phys.org/acp/4/1007/). Would the authors consider to investigate into this process?**

[R6] We agree with the reviewer that the uncertainties in the dry deposition of nitrate and ammonium can contribute to the model biases. However, the relatively

contributions of the dry deposition of nitrate and ammonium to the total deposition of nitrate+HNO$_3$ and ammonium+NH$_3$ is perhaps small (<10%) (Zhao et al., 2017). We expect a minor influence of such uncertainties on the SIA concentrations.

**A minor general comment: winter haze events are mentioned every now and then, without any clear context. Could the authors explain early in the paper why haze occurrence is an issue in the manuscript?**

[R7] We thank the reviewer for the suggestion and have added some descriptions about haze in Line 84-87 as follows: "*unusual biases of the meteorological fields and chemical processes may occur during the severe haze periods (daily mean PM$_{2.5}$ > 75 µg m$^{-3}$) (An et al., 2019). The models often significantly underestimate the PM$_{2.5}$ concentrations during the haze events (Wang et al., 2014; G. J. Zheng et al., 2015). Various model biases from meteorology, emissions, and the physical chemical processes interact with each other nonlinearly. It is therefore important to evaluate the model for all individual components of PM$_{2.5}$*".

**The comments and suggestions to the text:**
**Line 27. suggesting existing inaccuracies in the processes description (or presentation).**

[R8] We have revised this sentence accordingly.

**Lines 30-31 and 172-174. It's unclear what heterogeneous SO$_2$ oxidation reactions are in the model (by H$_2$O$_2$, ozone?)**

[R9] As stated in the Introduction, the mechanisms for the heterogeneous SO$_2$ oxidation is still under debate. We used non-mechanism-based parameterizations on RH or ALWC to simulate the heterogeneous sulfate formation.

**Lines 34-35. Again, can the author show that ISORROPIA is working properly? Clarify 'related to removal'. Wet or dry, or both.**

[R10] As discussed in [R5], the bias from ISORRPOPIA are expected to be minor compared with other potential biases. Because the chemical production, meterology, and the wet deposition cannot explain the model bias of nitrate, the removal here mainly mean dry deposition of HNO$_3$ and nitrate as well as the photolysis of particulate nitrate. We have revised the text.

**Line 42. what is considered to be reasonable?**

[R11] We have revised the statement as follows: "*have shown that the CTMs can reproduce the spatial and temporal variations of the surface PM2.5 concentrations in China*".

**Line 45. 'the model performance on PM$_{2.5}$ is component-dependent' sounds strange. Maybe like: even though the model represents well observed PM$_{2.5}$, it may happen due to compensation errors in model simulated PM$_{2.5}$ components.**

[R12] We have revised the text as follows: "*However, when the simulations of PM$_{2.5}$*

*components have compensating errors, the model still reproduces the PM$_{2.5}$ mass*".

**Line 48. I agree to some extend about SOA, but not about sulphate and nitrate (see my Ref. to Bian above). Perhaps the authors mean only specific studies for China.**
[R13] We have specified the region as follow: "*Recent model evaluations in China have reached an agreement that CTMs generally underestimate the concentrations of organic aerosol (OA) and sulfate but overestimate the concentrations of nitrate*".

**Line 57. Suggestion: The uncertainties in the emissions of primary PM and gaseous precursors of secondary PM are quite large.**
[R14] We have revised this sentence accordingly.

**Line 82. Suggested: therefore it is important to evaluate the model for all individual components of PM$_{2.5}$**
[R15] We have added this to the text in Line 86-87.

**Line 83. The measurements' artefacts can also decrease the discrepancies. . . the point is that in such cases, the model evaluation results give a wrong message.**
[R16] Yes, we agree. The statement in Line 88 has been revised as follows: "*On the other hand, observations may be biased, which is rarely considered when evaluating the model-observation discrepancies*".

**Line 103. Write: Institute of Atmospheric Physics (IAP) – for future use of abbreviation. Could you write here what site type is this (urban/suburban background?)**
[R17] Corrections have been made accordingly.

**Line 130. Suggestion: For comparison with observations at IAP. Beijing, the model simulations were performed for the ASCM measurements period.**
[R18] Corrections have been made accordingly.

**Line 132: Do I understand right that model simulations for 2012 meteorological conditions were used for comparison with 2006-2016 observations. Could the authors then say how (un)typical the 2012 weather was. Were year dependent emissions used, or also the same for 2012?**
[R19] Yes, the model simulations for 2012 meteorological conditions and emissions are used to comparison with 2006-2016 observations for computation efficiency. The meteorological conditions in 2012 is generally typical. Weather parameters like mean wind speed are in the middle range for 2006 to 2016 (Gao et al., 2020). The inter-annual variabilities of emission vary between species (Zheng et al., 2018; Li et al., 2019). For example, the emissions of SO$_2$ and primary PM$_{2.5}$ decreased by 50% and 40% from 2006 to 2016, respectively. The changes of NOx emissions are minor. But the NMVOC emissions increased by ~25%. Such changes have been considered in the evaluation of the model-observation discrepancies in Fig. 1. Because the measurements were mostly

conducted from 2011 to 2014 (47/77), the bias of using the fixed 2012 emissions on the general model evaluation is not evident.

**Line 150. Suggested: 80% which is considered to be a reasonable assumption (instead of 'for simplicity')**
[R20] We have deleted "*for simplicity*" in the text.

**Line 151. However, Hodzic. . . showed that the results were not very sensitive to. . ...**
[R21] We have revised this sentence accordingly.

**Line 161-162. What is the relative contribution of non-agricultural $NH_3$ emissions compared to the agricultural ones? This is important to know when analysing ammonium nitrate formation in cities.**
[R22] Nationally, the agriculture emissions contribute to 88.5% of total emissions of $NH_3$ (Zhang et al., 2018). Non-agricultural $NH_3$ emissions are important in urban areas. The contribution of non-agriculture $NH_3$ may reach 90% during haze periods in some places (Pan et al., 2016; Sun et al., 2017).

**Lines 172-180. Move up to Chemistry description, above the emissions.**
[R23] We have revised the manuscript accordingly.

**Line 182. Suggestion: Model performance for the individual PM components**
[R24] We have revised the subtitle as "Compensating errors from simulations of individual $PM_{2.5}$ components".

**Line 188: The modelled ammonium concentrations compare with observations better than simulated sulfate.**
[R25] We have revised the manuscript accordingly.

**Line 191. Further, we find that the model biases. . .**
[R26] This sentence has been deleted as suggested in the next comment.

**Lines 192 and 208-210 some repetition.**
[R27] We thank the reviewer for the suggestion and have reorganized the discussion about seasonality (see R8 in the response to Reviewer #1).

**Line 196. Would you expect the model performance to differ over China? Why? How different those regions are (weather, emissions?). Does that mean that the model is insensitive to the differences in meteo and chemical regimes?**
[R28] For general model problems like missing heterogeneous production, we expect similar model performance in different regions. If the model bias is related to localized issues like emissions, the model performance might be different among regions. Indeed, the purpose of evaluating the model results spatially is to help diagnosing.

**Line 208. Suggestion: On a seasonal basis,..**

[R29] We have revised this phrase accordingly in Line 201.

**Line 210-11. ..the overestimation of nitrate concentration is largest in spring, summer and autumn.., while the model bias is much smaller in winter. . .**

[R30] Corrections are made.

**Line 212-213. Are those simulations also with GEOS-Chem and ISORROPIA?**

[R31] Yes, simulations in both Wang et al. (2013) and Heald et al. (2012) are based on GEOS-Chem and ISORROPIA II.

**Line 214. In all seasons?**

[R32] Yes, the overestimation of nitrate occurs in all seasons. We have added "*in all seasons*" in the text.

**Line 217. The model performs worst in autumn**

[R33] We have revised the text accordingly.

**Line 218. This is about the only time when the correlation is mentioned. Why it's considered important here, but not for the other components? Is the relative importance of ASOA greater in summer?**

[R34] The correlation of summertime OA is mentioned because the $R$ value (0.28) is quite low compared with the values for other seasons ($\geq$0.49). The $R$ values are higher for other components and do not vary much by seasons (Table S3). The relative importance of ASOA is greater in winter than summer.

**Line 220. –compared to the 2 years of hourly observations. . ..**

[R35] We have revised the text accordingly.

**Line 223. ..underestimation of sulfate and OA by the overestimation of nitrate. . .**

[R36] We have revised the text accordingly.

**Line 226-27. Explain exclusion of the observations over 150 ug m$^{-3}$**

[R37] As explained in R3 and R17 of the response to Reviewer #1, the modeled sulfate, nitrate, ammonium, and OA are not specified to any size domain. In polluted environments, the modeled concentrations plausibly represent $PM_{2.5}$ not $PM_1$. The previous AMS or ACSM measurements are however for $PM_1$. Measurements in North China Plain suggest that 0.8 is a good coefficient to use for converting $PM_1$ to the $PM_{2.5}$ mass for these species. We therefore applied 0.8 when comparing the observations to the model results. The submicron-to-fine mass ratios may decrease to 0.5 during the severe haze episodes under high RH. The test here by excluding data over 150 $\mu$g m$^{-3}$ is to exclude the severe haze periods, and thus to prove that the discussion won't be affected by the submicron-to-fine ratio. For clarification, we have revised the text in Line 114-116 and Line 236-239.

**Line 242. Does that mean: If the evaporation of ammonium nitrate . . .. . .was accounted for in the model, the day time variation.. could be flatter?**

[R38] Yes. We have revised the sentences in Line xx. Instead, we would discuss more about the nitrate partitioning fraction.

**Line 254. Semivolatile POA scheme previously used? in GEOS-Chem**

[R39] Yes, the Simple SOA scheme is a new scheme in GEOS-Chem for SOA simulations.

**Line 267. . . .simulations lead**

[R40] The correction is made.

**Line 271. Suggestion: The uncertainties related with emission data including their temporal profiles) are considered to be one of the major sources of inaccuracies in modelled concentrations. . ...**

[R41] We have revised the text accordingly.

**Line 275. It is widely shown that regional models cannot accurately reproduce $NO_2$ at urban sites. Would the authors really expect the model with a resolution of 50-60 km to be capable of managing that?**

[R42] We agree with the reviewer. For a $0.25° \times 0.3125°$ horizontal resolution, $NO_2$ can generally disperse fully in the grid even in summer when the mean wind speed about 1.7 m/s and $NO_x$ lifetime about 5.9 h (Shah et al., 2020). We have clarified the impact of model grid size in Line 291. Good correlations of the simulated $NO_2$ concentrations at IAP site between $0.5° \times 0.625°$ and $0.25° \times 0.3125°$ horizontal resolutions ($R > 0.75$) suggest that the model with the resolution of 50-60 km still capture the variations of $NO_2$ concentrations, perhaps because Beijing is relatively big compared with other urban cities.

**Lines 282-285. What is the main sources and relative importance of non-agricultural $NH_3$? Why is it especially important during haze events?**

[R43] Non-agricultural $NH_3$ emissions are mainly from traffic, biofuel burning, chemical industry, and waste disposal (Kang et al., 2016). The increased contribution from non-agricultural $NH_3$ emissions in urban areas perhaps result from the limited transport of agricultural $NH_3$ emission from rural to urban during the haze periods under stagnant weather conditions (Pan et al., 2016). We have made this clear in Line 298-230.

**Lines 288-290: Unclear what is said here.**

[R44] We have revised this part in Line 304-307 as follows: "*The model also underestimates the aromatic VOC concentrations, similar to previous studies (Liu et al., 2012). Such underestimation would not affect the SOA simulations herein because that the Simple SOA scheme no longer derive aromatic SOA from the aromatic VOC*

*concentrations. Instead, the model treats aromatic SOA as a part of anthropogenic SOA, which is estimated on the basis of the parameterizations on CO*".

**Lines 311-12. Should be formulated more clear: The photolysis rate of particle-phase $HNO_3$ was shown in aged air masses to be higher than for the gaseous $HNO_3$. . ..., but in Beijing particulate $NO_3$ may have lower photolysis rates, because. . ...**
[R45] We have revised the text accordingly.

**Line 315. From the factors**
[R46] The correction is made.

**Line 319-20. . . .weeks were free of severe haze episodes (with extreme conditions which the model fails to reproduce????)**
[47] We have revised the text accordingly.

**Line 338. The increased wet deposition of nitrate. . .**
[R48] The correction is made.

**Line 360. Faster photolysis of particulate nitrate? Sounds contradictory to what is written on lines 311-312**
[R49] Photolysis of particulate nitrate would reduce the nitrate concentrations (in turn reduce $HNO_3$ concentrations by partitioning and increase NOx concentrations), which would lead to better model-observation agreements. We have revised this sentence in Line 380-384 as follows: "*Insufficient dry deposition of $HNO_3$ and nitrate and photolysis losses of particulate nitrate (i.e., to produce HONO and $NO_x$) in the model as well as the joint influence of multiple factors (discussed later) are possible explanations for the overestimation of nitrate*".

**References**

Bian, H. S., Chin, M., Hauglustaine, D. A., Schulz, M., Myhre, G., Bauer, S. E., Lund, M. T., Karydis, V. A., Kucsera, T. L., Pan, X. H., Pozzer, A., Skeie, R. B., Steenrod, S. D., Sudo, K., Tsigaridis, K., Tsimpidi, A. P., and Tsyro, S. G.: Investigation of global particulate nitrate from the AeroCom phase III experiment, Atmos. Chem. Phys., 17, 12911-12940, https://doi.org/10.5194/acp-17-12911-2017, 2017.

Chen, L., Gao, Y., Zhang, M., Fu, J. S., Zhu, J., Liao, H., Li, J., Huang, K., Ge, B., Wang, X., Lam, Y. F., Lin, C. Y., Itahashi, S., Nagashima, T., Kajino, M., Yamaji, K., Wang, Z., and Kurokawa, J.: MICS-Asia III: multi-model comparison and evaluation of aerosol over East Asia, Atmos. Chem. Phys., 19, 11911-11937, https://doi.org/10.5194/acp-19-11911-2019, 2019.

Gao, M., Liu, Z., Zheng, B., Ji, D., Sherman, P., Song, S., Xin, J., Liu, C., Wang, Y., Zhang, Q., Xing, J., Jiang, J., Wang, Z., Carmichael, G. R., and McElroy, M. B.: China's emission control strategies have suppressed unfavorable influences of climate on wintertime $PM_{2.5}$ concentrations in Beijing since 2002, Atmos. Chem. Phys., 20, 1497-1505, https://doi.org/10.5194/acp-20-1497-2020, 2020.

Heald, C. L., Collett, J. L., Lee, T., Benedict, K. B., Schwandner, F. M., Li, Y., Clarisse, L., Hurtmans, D. R., Van Damme, M., Clerbaux, C., Coheur, P.-F., Philip, S., Martin, R. V., and Pye, H. O. T.: Atmospheric ammonia and particulate inorganic nitrogen over the United States, Atmos. Chem. Phys., 12, 10295-10312, https://doi.org/10.5194/acp-12-10295-2012, 2012.

Kang, Y., Liu, M., Song, Y., Huang, X., Yao, H., Cai, X., Zhang, H., Kang, L., Liu, X. J., Yan, X., He, H., Zhang, Q., Shao, M., and Zhu, T.: High-resolution ammonia emissions inventories in China from 1980 to 2012, Atmos. Chem. Phys., 16, 2043-2058, https://doi.org/10.5194/acp-16-2043-2016, 2016.

Li, M., Zhang, Q., Zheng, B., Tong, D., Lei, Y., Liu, F., Hong, C., Kang, S., Yan, L., Zhang, Y., Bo, Y., Su, H., Cheng, Y., and He, K.: Persistent growth of anthropogenic non-methane volatile organic compound (NMVOC) emissions in China during 1990–2017: drivers, speciation and ozone formation potential, Atmos. Chem. Phys., 19, 8897-8913, https://doi.org/10.5194/acp-19-8897-2019, 2019.

Liu, M. X., Song, Y., Zhou, T., Xu, Z. Y., Yan, C. Q., Zheng, M., Wu, Z. J., Hu, M., Wu, Y. S., and Zhu, T.: Fine particle pH during severe haze episodes in northern China, Geophys. Res. Lett., 44, 5213-5221, https://doi.org/10.1002/2017gl073210, 2017.

Nenes, A., Pandis, S. N., Weber, R. J., and Russell, A.: Aerosol pH and liquid water content determine when particulate matter is sensitive to ammonia and nitrate availability, Atmos. Chem. Phys., 20, 3249-3258, https://doi.org/10.5194/acp-20-3249-2020, 2020.

Pan, Y., Tian, S., Liu, D., Fang, Y., Zhu, X., Zhang, Q., Zheng, B., Michalski, G., and Wang, Y.: Fossil Fuel Combustion-Related Emissions Dominate Atmospheric Ammonia Sources during Severe Haze Episodes: Evidence from [15]N-Stable Isotope in Size-Resolved Aerosol Ammonium, Environ. Sci. Technol., 50, 8049-8056, https://doi.org/10.1021/acs.est.6b00634, 2016.

Shah, V., Jacob, D. J., Li, K., Silvern, R. F., Zhai, S., Liu, M., Lin, J., and Zhang, Q.: Effect of changing $NO_x$ lifetime on the seasonality and long-term trends of satellite-observed tropospheric $NO_2$ columns

over China, Atmos. Chem. Phys., 20, 1483-1495, https://doi.org/10.5194/acp-20-1483-2020, 2020.

Sun, K., Tao, L., Miller, D. J., Pan, D., Golston, L. M., Zondlo, M. A., Griffin, R. J., Wallace, H. W., Leong, Y. J., Yang, M. M., Zhang, Y., Mauzerall, D. L., and Zhu, T.: Vehicle Emissions as an Important Urban Ammonia Source in the United States and China, Environ. Sci. Technol., 51, 2472-2481, https://doi.org/10.1021/acs.est.6b02805, 2017.

Wang, Y., Zhang, Q. Q., He, K., Zhang, Q., and Chai, L.: Sulfate-nitrate-ammonium aerosols over China: response to 2000-2015 emission changes of sulfur dioxide, nitrogen oxides, and ammonia, Atmos. Chem. Phys., 13, 2635-2652, https://doi.org/10.5194/acp-13-2635-2013, 2013.

Zhang, L., Chen, Y., Zhao, Y., Henze, D., Zhu, L., Song, Y., Paulot, F., Liu, X., Pan, Y., Lin, Y., and Huang, B.: Agricultural ammonia emissions in China: reconciling bottom-up and top-down estimates, Atmos. Chem. Phys., 18, 339-355, https://doi.org/10.5194/acp-18-339-2018, 2018.

Zhao, Y., Zhang, L., Chen, Y., Liu, X., Xu, W., Pan, Y., and Duan, L.: Atmospheric nitrogen deposition to China: A model analysis on nitrogen budget and critical load exceedance, Atmos. Environ., 153, 32-40, https://doi.org/10.1016/j.atmosenv.2017.01.018, 2017.

Zheng, B., Tong, D., Li, M., Liu, F., Hong, C. P., Geng, G. N., Li, H. Y., Li, X., Peng, L. Q., Qi, J., Yan, L., Zhang, Y. X., Zhao, H. Y., Zheng, Y. X., He, K. B., and Zhang, Q.: Trends in China's anthropogenic emissions since 2010 as the consequence of clean air actions, Atmos. Chem. Phys., 18, 14095-14111, https://doi.org/10.5194/acp-18-14095-2018, 2018.